

# Cascading transitions in the climate system

Mark Dekker[1,2,3], Anna S. von der Heydt[1,2], and Henk A. Dijkstra[1,2]

[1]Institute for Marine and Atmospheric research Utrecht, Department of Physics, Utrecht University, Utrecht, the Netherlands.
[2]Center for Complex Systems Studies, Utrecht University, Utrecht, the Netherlands.
[3]Department of Computer Science, Utrecht University, Utrecht, the Netherlands.

*Correspondence to:* Mark Dekker <m.m.dekker@uu.nl>

**Abstract.** We provide a theory of cascading tipping, i.e., a sequence of abrupt transitions occurring because a transition in one subsystem changes the background conditions for another subsystem. A mathematical framework of elementary deterministic cascading tipping points in autonomous dynamical systems is presented containing the double-fold, fold-Hopf, Hopf-fold and double-Hopf as most generic cases. Statistical indicators which can be used as early warning indicators of cascading tipping events in stochastic, non-stationary systems are suggested. The concept of cascading tipping is illustrated through a conceptual model of the coupled North Atlantic Ocean - El-Niño Southern Oscillation (ENSO) system, demonstrating the possibility of such cascading events in the climate system.

## 1 Introduction

Earth's climate system consists of several subsystems, e.g., the ocean, atmosphere, ice and land, which are coupled through fluxes of momentum, mass and heat. Each of these subsystems is characterised by specific processes, on very different time scales, determining the evolution of its observables. For example, processes in the atmosphere occur on much smaller time scales than in the ocean and hence in weather prediction, the upper ocean sets the background state for the evolution of the atmosphere. Similarly, in equatorial ocean-atmosphere dynamics associated with the El Niño - Southern Oscillation (ENSO) phenomenon, the global meridional overturning circulation can be considered a background state, as it evolves on a much larger time scale.

This notion that one subsystem provides a background state for the evolution of another subsystem is important when critical transitions are considered. In the climate system, a number of tipping elements have been identified (Lenton et al., 2008), where changes in observables can occur relatively rapidly compared to the changes in their forcing near so-called tipping points. Examples of tipping elements are the Atlantic Meridional Overturning Circulation (AMOC) (Stommel, 1961), the Arctic sea ice (Bathiany et al., 2016), monsoon patterns, midlatitude atmospheric flow (Barriopedro et al., 2006), vegetation cover (Hirota et al., 2011) and more local systems like coral reefs and permafrost. When one subsystem undergoes a transition, which changes the background state of another subsystem, also a transition may be induced in that second subsystem. Such dynamical interactions leading to coupled transitions are examples of 'tipping cascades' or 'domino effects' (Kriegler et al., 2009; Lenton and Williams, 2013).





Many tipping points have been analysed in separate subsystems, both for phenomena of the present-day climate (Lenton, 2011; Bathiany et al., 2016), as well as in past climates (such as the abrupt cooling of the Younger Dryas (Livina and Lenton, 2007) and the desertification of the Sahel region (Kutzbach et al., 1996)). However, less attention has been given to the interaction between transitions in different subsystems. For example, when the AMOC collapses, precipitation patterns may change

such that the equilibrium structure of the vegetation cover in the Amazon rainforest is shifted (Aleina et al., 2013). This may result in another transition, concerned with forest growth or dieback. Another example is the influence of the AMOC on the trade winds (through meridional sea surface temperature gradients), that in turn influence the amplitude of ENSO. In models, a collapse of the AMOC has been found to intensify ENSO (Lenton and Williams, 2013; Timmermann et al., 2007; Dong and Sutton, 2007), although there are also other effects that would weaken ENSO (Timmermann et al., 2005).

An example in past climates is the coupling between the ocean's overturning circulation and land ice. The rapid glaciation of the Antarctic continent around the Eocene-Oligocene boundary (34 Ma) is often explained in terms of a $CO_2$ threshold being reached that allows a major ice sheet to grow (DeConto and Pollard, 2003; Gasson et al., 2014). However, a two-step signal is found in the oxygen isotopic ratio, $\delta^{18}O$, which is attributed to a deep-sea temperature drop followed by the (slower) growth of the Antarctic Ice Sheet (AIS). One suggestion to explain the two-step transition is that the deep-sea temperature drop is related

to a change in the pattern of the global MOC (Tigchelaar et al., 2011). The ice sheet formation is then argued to be driven by decreasing atmospheric $CO_2$ (Pearson et al., 2009). This leads to the question whether a cascading tipping event did occur: The switch in MOC (first tipping) has led to the changes in the atmospheric $CO_2$ (e.g. Elsworth et al. (2017)) which caused the growth of the AIS (second tipping).

In the last few years, much work has been done to formulate statistical indicators and early warning signals of tipping points.

A system close to critical transition shows features of a 'critical slowing down' (Dakos et al., 2008; Scheffer et al., 2009; Kuehn, 2011). In the vicinity of the tipping point, the system slowly loses its ability to recover from small perturbations. This results in increased variance, autocorrelation and potentially also increased skewness and flickering (Scheffer et al., 2009). Various methods providing a specific scalar together with a threshold when approaching the transition have been suggested, such as degenerate fingerprinting (Held and Kleinen, 2004; Thompson and Sieber, 2011) and detrended fluctuation analysis

(DFA) (Peng et al., 1994; Livina and Lenton, 2007).

When considering cascading tipping points, the autocorrelation of two time series and their interaction needs to be analyzed simultaneously. Podnobik and Stanley (2008) proposed an altered form of DFA to assess the cross correlation between two non-stationary time series and called this method detrended cross-correlation analysis (DCCA). In the computation of the fluctuation function, they used cross-covariance instead of auto-covariance and fit this to a power law. This concept is further

extended by defining a coefficient $\rho_{DCCA}$ that accounts for the auto-covariance of the individual time series (Zhou, 2008; Yuan et al., 2015). However, no statistical analysis and indicators have yet been formulated for cascading tipping events.

In this paper, we provide a quantitative approach to cascading tipping events. We start with a mathematical framework to formulate elementary cascading tipping points (section 2). Next, we introduce statistical indicators and potential early warning indicators to analyse cascading transitions, and apply them to ensemble simulations of the elementary cascading tipping points



(section 3). Finally, we apply the new concepts to an example within the climate system: the potential cascading tipping mechanism between the AMOC and ENSO (section 4). We summarise and discuss our findings in section 5.

## 2 Mathematical framework for cascading tipping

In the climate system, tipping points are usually related to rapid transitions, where an observable in the climate system may
change abruptly in a relatively short time compared to changes in the forcing of the observable. Such rapid changes often involve transitions from one equilibrium state to another, which can often be explained with classical bifurcation theory for autonomous dynamical systems. These concepts can to a certain extent also be applied to non-autonomous systems (so-called slow-fast systems) when the time variation of parameters can be viewed as a slow external forcing (Kuehn, 2011). They form also the basics to understand phenomena as noise-induced tipping (Thompson and Sieber, 2011) and rate-dependent tipping
(Ashwin et al., 2012).

Here we focus on bifurcation-induced tipping points, and consider two types of bifurcations that are thought to be relevant to mechanisms of abrupt changes in the climate system; the back-to-back saddle-node bifurcation is often used to explain transitions between two co-existing equilibria (multi-stable systems), while the Hopf bifurcation can explain the appearance of oscillatory behaviour (Thompson and Stewart, 2002). In this view, abrupt change in the system appears as a consequence of a
15 parameter crossing a specific critical value at the bifurcation point.

A back-to-back saddle-node bifurcation generically occurs in physical systems (having bounded states) when one parameter is varying and the simplest dynamical system having such a bifurcation is described by

$$\frac{dx}{dt} = a_1 x^3 + a_2 x + \phi \tag{1}$$

where the $a_i$ ($i \in 1, 2$) are constants, $\phi$ is a parameter, $x$ is the state variable and $t$ is time. There are multiple equilibria in the
20 system if and only if $a_1 < 0$, $a_2 > 0$ and within the parameter interval $|\phi| < ((-4a_1^3 a_2^3)/(27a_1^4))^{1/2}$. In this case, saddle-node bifurcations occur at $\phi_c = \pm((-4a_1^3 a_2^3)/(27a_1^4))^{1/2}$.

A Hopf bifurcation also generically occurs in physical systems and the simplest dynamical system in which it occurs when one parameter is varied is described by

$$\frac{dx}{dt} = a_1 y + a_2 (\phi - (x^2 + y^2))x$$
$$\frac{dy}{dt} = b_1 x + b_2 (\phi - (x^2 + y^2))y \tag{2}$$

where again the $a_i, b_i, i = 1, 2$ are constants, $\phi$ is the parameter, $(x, y)$ is the state vector and $t$ is time. The state vector satisfying (2) reaches a stable periodic orbit if and only if $a_1 b_1 < 0$ and $\phi > 0$; the transition from steady to periodic occurs at $\phi = 0$.

There are two other bifurcations when one parameter is varied (the transcritical and pitchfork bifurcation) but they are non-generic because special conditions must hold (e.g. symmetry) and so these are not considered here. Using the back-to-back and Hopf bifurcations, cascading tipping can be viewed as a combination of two coupled subsystems, where each subsystem
undergoes one of these two types of bifurcations. The coupling introduces a direction of the cascade and we take account of this





by defining a *leading* system, which during its transition changes a parameter in the *following* system. The changing parameter in the following system then can induce the second transition. In the following, we discuss four types of cascading tipping in terms of combinations of saddle-node and Hopf bifurcations.

## 2.1 Double-fold cascade

The most intuitive system that has the potential to undergo a cascading tipping event is a system where both the leading and the following system have saddle-node bifurcations ('folds'). Analogous to the system of (1), a dynamical system containing a double fold cascade is then:

$$\begin{cases} \dfrac{dx}{dt} = a_1 x^3 + a_2 x + \phi \\ \dfrac{dy}{dt} = b_1 y^3 + b_2 y + \gamma(x) \end{cases} \tag{3}$$

where $x$ is the state vector of the leading system, $y$ that of the following system, $a_i, b_i$ ($i \in 1, 2$) are constants, and $\phi$ is a
parameter in the leading system. The key is here that the function $\gamma$, which serves as a parameter in the following system, depends on the leading system. The most simple coupling between the two systems is represented by $\gamma(x) = \gamma_1 + \gamma_2 x$.

When $\phi$ is changed moving through the bistable regime of the leading system the coupling moves the following system through its own bistable regime (see Table 1). Figure 1a shows the equilibria of the leading system for different values of $\phi$, showing the bistable regime in the center of the figure, embedded in the back-to-back saddle-node structure. The equilibrium
structure of the following system is displayed in Fig. 1e, as a function of $\phi$. Varying $\phi$ alters the state of the leading system, which through the coupling $\gamma$ affects the state of the following system. This results in the existence of four stable states in the following system in the bistable regime of the leading system; two per state of the leading system. The leading system's state acts as a background condition modulating the position of the following system's equilibria and therefore, in case of transition, may drastically reposition the equilibria of the following system. This is intuitively visible in Fig. 2a, where a time
series example of the dynamical system in (3) shows a cascading tipping event (parameters shown in Table 1). When the leading system (black) is forced (by changing $\phi$) to move from a bistable to a monostable regime, it transits towards a new equilibrium. During this transition, the following system (red) is affected such that it leaves the regime in which it had four possible equilibria and also transits to a different state.

## 2.2 Fold-Hopf cascade

The second type of cascading tipping event involves a saddle-node bifurcation in the leading system and a subsequent Hopf bifurcation in the following system. Using analogous notation as in (3), the simplest system that captures this so-called fold-Hopf cascade is

$$\begin{cases} \dfrac{dx}{dt} = a_1 x^3 + a_2 x + \phi \\ \dfrac{dy}{dt} = b_1 z + b_2 (\gamma(x) - (y^2 + z^2)) y \\ \dfrac{dz}{dt} = c_1 y + c_2 (\gamma(x) - (y^2 + z^2)) z \end{cases} \tag{4}$$





where $x$ is again the state vector of the leading system, and $(y, z)$ that of the following system. By slowly varying the parameter $\phi$ (e.g., linearly as $\phi(t)$) the leading system moves through its bistable regime (see Tab. 1 for parameter values) and via the coupling $\gamma(x) = \gamma_1 + \gamma_2 x$ forces the following system across the Hopf bifurcation point.

    The bifurcation structure of the leading system of (4), using parameters stated in Table 1, is displayed in Fig. 1b. This (as

in Fig. 1a) shows the back-to-back saddle-node structure, a Hopf bifurcation and a subsequent oscillatory regime on part of the upper branch. The actual oscillation occurs in the following system, shown in Fig. 1f. On the lower branch of the leading system, the following system does not oscillate, but on the upper branch, for many values of $\phi$, the following system does. This makes it possible for steady and oscillatory states to coexist on the right side of the Hopf bifurcation in Fig. 1f. An example of a time series showing a fold-Hopf cascading event is shown in Fig. 2b. A transition in the leading system (black) brings the

following system (red/orange) into an unstable equilibrium that eventually leads to an oscillatory state.

## 2.3 Hopf-fold cascade

A third type of cascading event involves a Hopf bifurcation in the leading system and a subsequent saddle-node bifurcation in the following system. Using a similar notation as in the previous subsection, the simplest system with a Hopf-fold cascade (see Table 1 for parameter values) is given by

$$
\quad
\begin{cases}
\dfrac{dx}{dt} = a_1 y + a_2(\phi - (x^2 + y^2))x \\[2mm]
\dfrac{dy}{dt} = b_1 x + b_2(\phi - (x^2 + y^2))y \\[2mm]
\dfrac{dz}{dt} = c_1 z^3 + c_2 z + \gamma(x)
\end{cases}
\tag{5}
$$

where $(x, y)$ is the state vector of the leading system, and $z$ that of the following system. Again, we can slowly increase $\phi$ such that the leading system $(x, y)$ crosses a Hopf bifurcation; via the coupling $\gamma(x) = \gamma_1 + \gamma_2 x$ the following system is then moved through its bistable regime such that a fold is reached in $z$.

    Fig. 1c contains the typical bifurcation structure of the leading system in (5), containing a Hopf bifurcation separating

stationary from oscillatory behavior. The following system's equilibrium structure for varying $\phi$ is given by Fig. 1g. In this particular configuration, for any negative $\phi$ there are multiple stable equilibria. This makes sense, as $\phi$ only affects the following system via its impact on the leading system, and for negative $\phi$ the leading system remains constant. At $\phi = 0$, the Hopf bifurcation in the leading system is reached and $(x, y)$ start oscillating. The following system oscillates a little along with the leading system due to the oscillatory changing value of $\gamma$.

When $\phi$ increases more, the amplitude of the leading system's oscillation grows, which may make $\gamma$ to cross the threshold such that the following system leaves its bistable regime (be it temporarily as $\gamma$ will be reduced again due to the oscillation). This forces the following system into its upper branch, as can be seen in Fig. 1g by the red dashed lines in the lower branches ending at $\phi \approx 0.5$. The upper branch's displayed stable oscillation ends at $\phi \approx 0.8$, because the amplitude becomes large enough for the system to swap between multiple equilibria. An example of such a cascading transition event can be seen in Fig. 2c,

where an oscillation starts in the leading system (black/grey), of which a particular phase makes the following system (red) transit into the second equilibrium.





## 2.4 Double-Hopf cascade

A fourth type of cascading tipping event discussed here involves a Hopf bifurcation in the leading system and a subsequent Hopf bifurcation in the following system. Using analogous notation as above, this double Hopf cascade is captured by the dynamical system

$$
\begin{cases}
\dfrac{dx}{dt} = a_1 y + a_2(\phi - (x^2 + y^2))x \\[2mm]
\dfrac{dy}{dt} = b_1 x + b_2(\phi - (x^2 + y^2))y \\[2mm]
\dfrac{du}{dt} = c_1 v + c_2(\gamma(x) - (u^2 + v^2))u \\[2mm]
\dfrac{dv}{dt} = d_1 u + d_2(\gamma(x) - (u^2 + v^2))v
\end{cases}
\tag{6}
$$

where $(x, y)$ is the state vector of the leading system, and $(u, v)$ that of the following system. If $\phi$ forces $(x, y)$ such that it crosses the Hopf bifurcation point, the coupling $\gamma(x) = \gamma_1 + \gamma_2 x$ causes a crossing of the second Hopf bifurcation in $(u, v)$.

Figure 1d shows the bifurcation diagram of the leading system, showing a typical system with a Hopf bifurcation. The following system (Fig. 1h) is stationary, up to the point that the term $\gamma$ becomes high enough to start an oscillation. However, $\gamma$ oscillates with the leading system (for $\phi > 0$). This means that only in a particular part of the leading system's oscillation period, oscillatory behavior can be expected in the following system. This interaction between the two oscillations result in torus bifurcations for particular values of $\phi$. An example of a time series showing a Hopf-Hopf cascading transition is displayed in Fig. 2d. After a (slow) oscillation in the leading system (black/grey) has started, a (fast) oscillation in the following system (red/orange) arises in particular phases of the slow oscillation.

## 3 Early warning signals of cascading tipping points

In the previous section we have formulated elementary deterministic dynamical systems that can exhibit cascading tipping. In order to detect tipping events from e.g. observed time series in real systems, we need to detect whether a system is close to critical transition. In general, a system close to critical transition recovers more slowly from perturbations, which in turn increases memory in the time series. This leads to the phenomenon of 'critical slowing down' prior to bifurcation points. In this section, we present statistical indicators which may be useful to detect cascading tipping events.

### 3.1 Methods for single tipping points

Several methods have been suggested for the analysis of time series to detect the approach of a single tipping point. For saddle-node bifurcations, the key features of such a time series is a critical slowing down. This can be investigated as standard quantities such as increasing autocorrelation (e.g., the lag-1 autocorrelation), increasing variance and increasing skewness (Held and Kleinen, 2004; Scheffer et al., 2009; Kuehn, 2011). Although critical slowing down near critical transitions is a more general feature of (even chaotic) dynamical systems (Tantet et al., 2018), the standard quantities may not always provide an early warning of a critical transition. Hence, more complicated indicators have been introduced, such as (i) the degenerate





fingerprinting (DF) and (ii) the detrended fluctuation analysis (DFA), (Held and Kleinen, 2004; Thompson and Sieber, 2011; Peng et al., 1994; Livina and Lenton, 2007).

As critical slowing down implies an increasing autoregressive behavior in the time series prior to a transition, the memory component is increased. After time-equidistant interpolation and detrending of the data, in the DF method, one fits the following general autoregressive process to the time series $x_n$:

$$x_{n+1} = c \cdot x_n + \sigma \eta_n \tag{7}$$

where $\eta_n$ is Gaussian white noise and $c = \exp(-\lambda \Delta t)$ the AR(1) coefficient. Here $\lambda$ can be seen as the decay rate of perturbations in previous time steps. As the approaching of a bifurcation point involves an increase in memory, the value of $c$ is presumed to increase towards one when approaching a saddle-node bifurcation point.

DFA copes well with non-stationarity in time series while searching for long-range correlations (Peng et al., 1994; Livina and Lenton, 2007). In DFA, one first chooses an integer window size $s$ and divides the (cumulative-summed) time series $X(n)$ in $N_s = N/s$ segments that do not overlap, where $N$ is the length of the time series. In every window, the best polynomial fit of a chosen order is calculated. A quadratic polynomial is used here. The squared deviation from this quadratic polynomial for every window is summed, resulting in a measure of the auto-covariance fluctuating around the fit:

$$F^2(\nu, s) \quad = \quad \frac{1}{s} \sum_{i=1}^{s} [X((\nu-1)s+i) - x_\nu(i)]^2 \tag{8}$$

with $X$ the detrended time series and $x_\nu$ the best polynomial fit in segment $\nu$. Then, an average is taken over all segments to obtain the fluctuation function $F(s)$:

$$F(s) \quad = \quad \sqrt{\frac{1}{N/s} \sum_{\nu=1}^{N/s} F^2(\nu, s)} \tag{9}$$

which depends solely on $s$. The long-range auto-correlations can now be recognized by fitting the fluctuation function to a power-law and looking at the resulting DFA-exponent $\alpha$, according to

$$F(s) \quad \propto \quad s^\alpha \tag{10}$$

For $\alpha \leq 0.5$, there is no long-term correlation and the fluctuations are indistinguishable from white noise. However, when $\alpha > 0.5$, there are long-term correlations present and for $\alpha \geq 1.5$ the system has reached a bifurcation point. In the simulations analysed here the DFA scaling exponent is fitted explicitly for every (moving) window.

## 3.2 Detrended cross-correlation analysis

Cascading tipping involves two systems with their own bifurcation structure and their proximity towards bifurcation points. Although the leading system may be close to tipping, the following system might still be far away from its bifurcation point and needs the critical transition of the leading system to even come close to this point. This is why the general measures for single tipping events cannot be used, nor can regular cross (Pearson) correlation. The reason is that the following and leading system do not have a one-to-one relationship, but are rather coupled through specific parameters, only seen in long-range correlations.





When approaching a cascading tipping point, the long-range cross-correlation between the two state vectors (of the leading, say $x$, and following system, say $y$) is expected to increase. The state vector $x$ becomes more auto-correlated and is less susceptible to noise, and therefore through the coupling influences $y$ in a more robust way. To find long-range *cross*-correlations, a method so-called detrended cross-correlation analysis (DCCA) was developed (Zebende, 2011; Podnobik and Stanley, 2008;

Zhang et al., 2001; Zhou, 2008). Instead of taking the auto-covariance (8) to calculate the fluctuation function, one takes the cross-covariance,

$$F^2_{DCCA}(\nu, s) = \frac{1}{s} \sum_{i=1}^{s} [(X((\nu-1)s+i) - x_\nu(i)) \\ \cdot (Y((\nu-1)s+i) - y_\nu(i))]^2$$ (11)

with symbols similar to(8). With this function, one can calculate the fluctuation function and subsequent power-law scaling coefficient (Podnobik and Stanley, 2008; Zhang et al., 2001), similar to (9).

A variation on this method was proposed by Zebende (2011) and involves the ratio between $F^2_{DCCA}$ and $F_{DFA}$ of the two systems. Specifically, one chooses a certain segment size $s$ and computes:

$$\rho_{DCCA} = \frac{F^2_{DCCA}}{F_{DFA\{x\}} F_{DFA\{y\}}}$$ (12)

which measures the level of the long-term cross-correlation between variable $x$ and $y$; the quantity $\rho_{DCCA}$ has values between -1 and 1.

## 3.3   Analysis of cascading tipping systems

The previous section presented various quantities to analyse the occurrence of cascading tipping events. This section is devoted to give insight into the accuracy and the usefulness of the indicators, by applying them to the double-fold and fold-hopf cascading tipping cases.

### 3.3.1   Double-fold cascading tipping

To simulate these events and use statistical indicators, noise has to be included. The system of equations used here is:

$$\begin{cases} \dfrac{dx}{dt} = a_1 x^3 + a_2 x + \phi + \zeta_x \\ \dfrac{dy}{dt} = b_1 y^3 + b_2 y + \gamma(x) + \zeta_y \end{cases}$$ (13)

where now in addition to (3), $\zeta_x, \zeta_y$ are Gaussian white noise terms. We simulate an ensemble of 100 members with the parameter settings and initial conditions as displayed in Table 2. The results of this ensemble are displayed in Fig. 3. Running windows containing the transition itself are shaded white because this data is misleading when one wants to know what happens

before the bifurcation points. We make the distinction between the leading-transitional period (LTP), which is the time series before the tipping point in the leading system, and the following-transitional period (FTP), which is the time series between the first tipping point and the tipping point in the following system.





In the LTP, we can clearly see the gradual increasing leading system's variance, AR(1) coefficient and DFA scaling co-efficient. These are all evidence of the leading system slowly approaching a bifurcation point, according to the change in the parameter. There is not much evidence of long-range auto-correlations in the time series of the following system, as its variance is low and the DFA scaling exponent remains below 0.5, pointing towards that the detrended fluctuations are statistically white

noise. The AR(1) coefficient of the following system does increase just prior to the first tipping, but also stays small (compared to unity).

The detrended cross correlation scaling exponent (abbreviated here as DXA) does give $> 0.5$ values, but the range throughout the ensemble members is most of the time too large to see any structural development when approaching the bifurcation point. During the leading system's transition, a strong increase is visible, pointing towards the rather strong cross-correlated behavior

during this period (as the following system also shifts its equilibrium a little).

The quantity $\rho_{DCCA}$ seems to attain a small positive value (around 0.3) and stays relatively constant throughout the whole time interval. One important aspect of the calculation of $\rho_{DCCA}$, as we found by experimentation, is that the values is very sensitive to the segment size $s$ and the moving window size. The moving window determines the amount of data that is available to find long-range correlations, and the segment size has a strong impact on the accuracy of the fits and therefore

on the segmented fluctuations. As in this type of problem, we need a temporal evolution of the statistical indicators, we need moving windows and thus encounter this problem. As these indicators (DXA and $\rho_{DCCA}$) have been applied successfully in simpler systems (Zebende, 2011; Podnobik and Stanley, 2008; Zhang et al., 2001; Zhou, 2008), more research on the sensitivity of the indicators with respect to the segment size and moving window size may lead to more robust results.

During the FTP, the variance, AR(1) and DFA of the leading system are strongly reduced. However, the gradual increasing

of the following system's variance, AR(1) coefficient and DFA scaling coefficient are definitely visible, pointing towards the approaching of a bifurcation in the following system. Also notable is the contrast in the DFA of the following system between before and after the tipping of $x$. The DFA of $y$ went from a white-noise regime (around 0.5) before the tipping of $x$ towards a regime where the detrended fluctuations point towards long-range auto-correlations after the tipping of $x$ (1-1.5). This illustrates the relation between the leading system's state and the following system's DFA scaling exponent. The DXA remains relatively

high, but overall no structural development can be seen in this graph. The quantity $\rho_{DCCA}$ exhibits the same behavior as in the LTP, probably for the reasons already mentioned.

To assess the effects of the cascading effect on the mentioned statistics, we compare the results with a case where the system (13) does not undergo a tipping in the following system (so only one transition remains). The resulting ensemble results are shown in Fig. 4. The most important differences between the regular cascading event and a single tipping event can be found

when comparing the variance, AR(1) and DFA scaling coefficient changes between LTP and FTP (or period after the first transition). During the LTP, the leading system is close to transition and therefore has strong autoregressive behavior, which is the opposite for the following system, being far from its bifurcation point. During the FTP, the following system generally gains memory because it is brought closer to its transition point, and the leading system loses this because it had just arrived at a new state. So we expect that from the LTP towards the FTP, the variance, AR(1) and DFA *decrease* in the leading system,

and *increase* in the following system.





To quantify this effect, Table 3 shows the ratios of the different quantities, indicated by $Q$, during the FTP and LTP phases ($\bar{Q}_{FTP}/\bar{Q}_{LTP}$), for the cases *with* a second tipping (corresponding to runs shown in Fig. 3) and *without* second tipping (Fig. 4). All ensemble members are included in these numbers, accounting for a mean and standard deviation of these ratios. As expected, the leading system's autoregressive metrics decrease in both cases, visible in the mean values of the ratios of the leading system's autoregressive variables being lower than 1. Also as expected, the following system's autoregressive behavior increases (ratios $> 1$) in both cases, but striking is that in the case of a cascading tipping event (*with* second tipping), the following system's ratios are much higher than those in the case of a single tipping event (*without* second tipping). To investigate whether the difference in ratios between single or double tipping is indeed significant, a Student's t-test is done. The results are shown in Tab. 4. The high $p$-values for the leading system's ratios indicate no significant difference between single or double tipping, but the low $p$-values for the following system's ratios indicate a significant difference. This shows the potential of using the ratio of autoregressive variables before and after a transition to assess whether a cascading transition may follow. Further research is needed to quantify this expectation and to assess the sensitivity of these ratios to the system's parameters.

### 3.3.2 Fold-Hopf cascading tipping

Many statistical indicators have been applied on fold bifurcations specifically, because these transitions show a clear sign of critical slowing down and increased autocorrelation due to the irreversibility and process of going from one equilibrium towards another. A supercritical Hopf bifurcation has a different nature with respect to the slowing down, as it is no critical transition. We will now consider the fold-Hopf cascade in the light of the statistical indicators described before. For this, we use the following stochastic dynamical system:

$$
\begin{cases}
\dfrac{dx}{dt} = a_1 x^3 + a_2 x + \phi + \zeta_x \\
\dfrac{dy}{dt} = b_1 z + b_2(\kappa(x) - (y^2 + z^2))y + \zeta_y \\
\dfrac{dz}{dt} = c_1 y + c_2(\kappa(x) - (y^2 + z^2))z + \zeta_z
\end{cases}
\tag{14}
$$

similar to (4) but now white noise is added through the terms $\zeta_x, \zeta_y$ and $\zeta_z$. We used an ensemble of 100 simulations with the parameter settings and initial conditions as displayed in Table 2. The results of the ensemble are shown in Fig. 5. Here, we do not make the distinction between the LTP and the FTP, because in contrast to the double-fold cascade, the following system undergoes a transition that is easily reversed and the system either is stationary or oscillating. Noise directly starts the oscillation and completely removes the FTP. The following system is quickly drawn towards the equilibrium state $(x, y) = (0, 0)$, and the leading system is in a steady state. During the time towards the bifurcation point, the variance, AR(1) coefficient and DFA of the leading system $z$ gradually increase, as is expected as we force this system towards its bifurcation point.

The DFA and AR(1) of the following system after the bifurcation are in strong contrast with before the bifurcation, probably due to the autoregressive nature of the oscillation. The relation between the leading system's state and the following system's DFA scaling exponent is also confirmed in this case. The DXA sharply increased just prior to the critical transition but throughout the whole time series, retains relatively high values. The reason behind this might be found in the low level of noise that is



taken, or other simulation-specific parameters. It could also be that it is because the following system on average has a high, weakly varying DFA scaling exponent on itself, which in turn might affect the height and variability in the cross-correlation. The $\rho_{DCCA}$ coefficient remains positive and small, just like in the double-fold case. Again, this may have to do with the choice of window and segment sizes.

## 4  The coupled AMOC-ENSO system

In this section, the theory of cascading tipping will be applied to investigate the coupling between the Atlantic Meridional Overturning Circulation (AMOC) and the El-Niño -Southern Oscillation (ENSO).

### 4.1  Coupling between AMOC and ENSO

First, to demonstrate the coupling between AMOC and ENSO, we use output from global climate model simulations. In the ESSENCE project (Ensemble SimulationS of Extreme weather events under Nonlinear Climate changE) several simulations were performed with the ECHAM5/MPI-OM coupled climate model, including so-called hosing experiments (Sterl et al., 2008), where fresh water is added around Greenland to mimic ice sheet melting.

Of these climate model simulations we have used two ensembles; the first is the 'standard' experiment, where greenhouse gases evolve according to observations and from the year 2000 onwards following the SRES-A1b scenario (experiment name SRES-A1b). The second ensemble is the same as the standard one but has additional freshwater input (1 Sv = $10^6$ m$^3$/s) around Greenland from the end of year 2000 onwards (experiment name HOSING-1). The HOSING-1 ensemble contains a hosing experiment in the classical way, following the procedure of Jungclaus et al. (2006). Five runs of each ensemble are taken, specifically runs 041-045 of the HOSING-1 and runs 021-025 of the SRES-A1b ensemble. The temporal resolution used is monthly data between 1950 and 2100. The spatial fields are on a curvilinear grid, with 40 vertical levels in the ocean. We use deseasonalised data because we are interested in interannual variability, not in seasonal variability, as El-Niño is associated with these timescales. As an AMOC index, we use the maximum of the Atlantic meridional overturning stream function at 35°N and as ENSO index, we use the NINO3.4 index, which is the average SST over the region 170°W-120°W × 5°S - 5°N.

The results for the evolution of the AMOC and ENSO are shown in Fig. 6. It is clearly visible that the AMOC decreases strongly in the hosing experiments, by approximately 85%. Table 5 compares statistical properties for the time interval before and after 2001 (which is the year at which the hosing starts). We use the non-anomaly statistics, as this gives us information about the differences in the mean. We do note that we only use five runs per ensemble, which makes the uncertainty not statistically robust. We only state it in Table 5 to give an idea of the range of the variables among the different runs.

It is visible in Table 5 that the variability of NINO3.4 increases (bold numbers) if we compare the periods of 1950-2000 and 2001-2100. This increased variability is visible in both the standard and the HOSING-1 runs. However, the variability is increased much stronger in the HOSING-1 experiment, indicating that the decrease of the AMOC indeed has an amplifying effect on ENSO. The large difference between the standard and hosing runs suggests that the NINO3.4 index changed in the hosing experiment, as a consequence of the decrease of the AMOC.





Several mechanisms have been suggested in the literature on the coupling between the AMOC and ENSO. The first mechanism is concerned with oceanic waves. A colder North Atlantic creates density anomalies that induces oceanic Kelvin waves to propagate southward (along the American coast) across the equator. In West Africa, this energy radiates as Rossby waves towards the north and south, which induces Kelvin waves to move along the tip of south Africa into the Indian ocean, that

eventually reach the Pacific. Consequently, the eastern equatorial Pacific thermocline deepens on a timescale of decades. This deepening has a weakening effect on the amplitude of ENSO (Timmermann et al., 2005).

The second mechanism is concerned with the trade winds. Cooling in the northern tropical Atlantic (due to AMOC weakening) induces anti-cyclonic atmospheric circulation (Xie et al., 2007) that intensifies the northerly trade winds over the northeastern tropical Pacific. This leads to a southward displacement of the Pacific ITCZ (Zhang and Delworth, 2005) and this

generates a meridional SST anomaly due to anomalous heat transport and the wind-evaporation SST feedback in the Pacific. Also, Dong and Sutton (2007) found an atmospheric coupling through Rossby waves sent into the northeast tropical Pacific. The result of the wind stress as coupling between the two systems is an intensification of ENSO and this mechanism is argued to be stronger than the coupling through oceanic waves (Timmermann et al., 2005).

## 4.2    A coupled AMOC - ENSO model

To study the possible cascading transition through the wind-stress coupling, we use a conceptual model. For the AMOC, the classical Stommel box model (Stommel, 1961) is used. It consists of a polar (subscript $p$) and an equatorial box (subscript $e$), both with a temperature $T$ and salinity $S$ and coupled by a density-driven flow rate. The state variables are then defined as $\Delta T = T_e - T_p$ and $\Delta S = S_e - S_p$. The time evolution of these variables is as follows (Cessi, 1994):

$$\begin{cases} \dfrac{d\Delta T}{dt} = & -\dfrac{1}{t_r}(\Delta T - \theta_0) - Q(\Delta\rho)\Delta T \\ \dfrac{d\Delta S}{dt} = & \dfrac{F_s}{H}S_0 - Q(\Delta\rho)\Delta S \end{cases} \tag{15}$$

where the first term in the temperature equation refers to relaxation towards a background temperature, and the second term refers to density-driven meridional transport. Specifically, $t_r$ is the surface temperature restoring time scale and $\theta_0$ is the equator-to-pole atmospheric temperature difference. $Q(\Delta\rho)$ is the transport function, which is calculated from a diffusion time scale and the meridional density gradient $\Delta\rho$. In the salinity equation, $S_0$ is a reference salinity, and $H$ is the ocean depth. The parameter $F_s$ is the freshwater flux, which can be used as a bifurcation parameter. The streamfunction

$\Psi = \gamma_0 \Delta\rho / \rho_0 = \gamma_0 (\alpha_T \Delta T - \alpha_s \Delta S)$

represents the strength of the AMOC, with $\gamma_0 > 0$ a flow constant, $\rho_0$ a reference density and $\alpha_T, \alpha_S$ the thermal and haline expansion/contraction coefficients.

For the El-Niño Southern Oscillation, we use the conceptual model as proposed in Timmermann et al. (2003). This model has a state vector consisting of the temperature of the western Pacific $T_1$, the temperature of the eastern Pacific $T_2$ and the

thermocline depth of the western Pacific $h_1$. The model finds its basis in the Zebiak and Cane (1987) ENSO model, with a





two-strip and two-box approximation, and a shallow-water model for the upper ocean with a fixed mixed layer depth:

$$
\begin{cases}
\dfrac{dT_1}{dt} = & -\alpha(T_1 - T_r) - \dfrac{u(T_2 - T_1)}{L/2} \\
\dfrac{dT_2}{dt} = & -\alpha(T_2 - T_r) - \dfrac{w(T_2 - T_{sub})}{H_m}
\end{cases}
\tag{16}
$$

with $1/\alpha$ a typical thermal damping timescale, $T_{sub}$ the temperature below the mixed layer, $H_m$ and $L$ the depths of the mixed layer and basin width, respectively, $w$ upwelling velocity and $u$ atmospheric zonal surface wind being linear to wind stress: $u/(L/2) = \epsilon\beta\tau$ and $w/H_m = -\zeta\beta\tau$. The parameters $\epsilon$ and $\zeta$ refer to the strength of zonal and vertical advection (bifurcation parameters).

The subsurface temperature $T_{sub}$ is parametrized as

$$
T_{sub} = T_r - \frac{T_r - T_{r0}}{2}\left[1 - \frac{\tanh(H + h_2 - z_0)}{h*}\right]
\tag{17}
$$

with $h_2$ the east equatorial Pacific thermocline depth (calculated as deviation from a reference depth $H$), $z_0$ the depth for which $w$ becomes its characteristic value and $h*$ the sharpness of the thermocline. The thermocline depths are calculated as follows:

$$
\begin{cases}
h_2 = & h_1 + bL\tau \\
\dfrac{dh_1}{dt} = & r\left(-h_1 - \dfrac{bL\tau}{2}\right)
\end{cases}
\tag{18}
$$

where $b$ the efficiency of wind stress $\tau$ to drive the thermocline tilt. For further details and parameter values, we refer to Timmermann et al. (2003).

The coupling of the AMOC and ENSO systems is mainly through influence on the wind stress. In the original model, the zonal wind stress $\tau$ is expressed as:

$$
\tau = \frac{\mu(T_2 - T_1)}{\beta}
\tag{19}
$$

with $\mu/\beta$ parameters that control the influence of the zonal temperature gradient on the wind stress, set to be $0.02$ Pa·K$^{-1}$. However, in Dijkstra and Neelin (1995) it was argued that part of the contribution to the zonal wind stress, $\tau_{ext}$, arises from processes outside the tropical Pacific. Here, we model $\tau_{ext}$ to be dependent on the meridional temperature gradient in the Atlantic $\Delta T$, i.e.,

$$
\tau = \tau_{ext}(\Delta T) + \frac{\mu}{\beta}(T_2 - T_r)
\tag{20}
$$

with a negative relation between $\tau_{ext}$ and Atlantic meridional SST gradient $\Delta T$ as we know from literature described above (stronger positive $\Delta T$ results in stronger easterlies, thus negative $\tau_{ext}$). Note that both the total Pacific wind stress $\tau$ and specifically $\tau_{ext}$ should always be negative. The total wind stress is negative because this area (at low altitude) is strictly dominated by easterly winds, and $\tau_{ext}$ is negative because through the meridional temperature gradient, it reflects the influence of the zonal mean Hadley cell on the equatorial Pacific. Physically, the Hadley cell only induces negative zonal wind stress in this region.





In the coupling (20), we fix $\beta$ and vary $\mu$ as the coupling parameter. For $\tau_{ext}$ we take a linear relation:

$$\tau_{ext} = \alpha_\tau \Delta T + \gamma_\tau - \tau_0 \qquad (21)$$

where all coefficients are constant over time. The parameters $\alpha_\tau$ and $\gamma_\tau$ can be estimated from the ESSENCE data as discussed in section 4.1, and the parameter $\tau_0$ is there to remove a constant part in the zonal mean wind stress because we are interested

in the contribution of changes in the meridional overturning. Using five ESSENCE runs per ensemble for both the standard forcing and hosing-experiment, respectively, $\Delta T$ is computed as the absolute difference between the SST in the North Atlantic area $(50-60°\text{N} \times 50-20°\text{W})$ and the Equatorial Atlantic region $(0-20°\text{N} \times 45-20°\text{W})$. For the wind stress $\tau_{ext}$, the zonally integrated wind stress averaged over the region $0-10°\text{N}$ is taken. In Fig. 7, 5-year running means of annual averages are plotted for the hosing simulations (in red) and the standard simulations (in black). Clearly, $\tau_{ext}$ decreases with increasing $\Delta T$, such

that when the AMOC collapses (larger $\Delta T$) the wind stress $\tau_{ext}$ becomes more negative and the external part of the trade winds increases. However, we also note that the spread in the simulation data is large, which in part can be attributed to internal variability present in the simulations. The coefficients $\alpha_\tau$ and $\gamma_\tau$ were found to be (from a linear fit) -0.000376 Pa·°C$^{-1}$ and -0.0119 Pa. By looking at the $\Delta T$ regime in Fig. 7, $\tau_0$ is chosen to be the wind stress at 19 °C: $\tau_0 = \alpha_\tau \cdot 19 + \gamma_\tau \approx -0.0190$ Pa. This results in a final quantized expression for the coupling:

$$\tau_{ext} \approx -0.000376 \cdot \Delta T + 0.00715 \qquad (22)$$

## 4.3 Results

The AMOC model's bifurcation diagrams are shown in Figs. 8a and b, clearly showing a back-to-back saddle-node structure. For an interval of values of the freshwater flux $F_s$, the system has multiple equilibria, and for other values, only one equilibrium remains. This means that when we are in the high-$\Psi$ branch and $F_s$ is large enough, the system can make a transition to the

20 low-$\Psi$ branch. This is depicted by the blue arrow in Fig. 8b.

The bifurcation diagram of the ENSO model with $\tau_{ext}$ as parameter is shown in Fig. 8c. First of all, the bifurcation diagrams become much simpler than in the original Timmermann et al. (2003) model, the reason for this being extensively discussed in Dijkstra and Neelin (1995). Fig. 8d shows the influence of $\mu$ for the position of the oscillatory regime: on each branch, two Hopf bifurcations can be found and the $\mu$ value of the first Hopf bifurcation decreases with more negative $\tau_{ext}$. This indicates

that the El-Niño intensifies when the easterly external wind is increased.

Using $\tau_{ext}$ to couple the AMOC and ENSO models, we performed simulations with $\Delta t = 0.1$ and the Runge-Kutta fourth order integration method. To initiate the collapse of the overturning, a freshwater forcing $F_s$ is applied in the form of a step function:

$$F_s = \begin{cases} 0.006 & \text{if } t \le 500 \text{ y} \\ 0.01 & \text{if } t > 500 \text{ y} \end{cases} \qquad (23)$$

Using the coupling of Eqn. 22, we attain the results shown in Fig. 9. The exact quantification of this partly modulates which effect the collapse of the AMOC has on ENSO. For the chosen coupling, the collapse of the overturning leads to the crossing



of the first Hopf bifurcation point in the following system, and an oscillation starts growing. Hence, this is a typical illustration of the fold-Hopf cascading behavior discussed above.

## 5 Summary, Discussion and Conclusions

In this paper, we introduced the concept of cascading tipping, which can occur when a transition in a leading system alters
background conditions for a following system such that it also undergoes a transition. We presented a mathematical framework around this concept, where we used generic bifurcations (back-to-back saddle-node and Hopf) in both leading and following systems. Four types of deterministic dynamical systems with the possibility for cascading events were formulated, including the double-fold cascade, the fold-Hopf cascade, the Hopf-fold cascade and the double-Hopf cascade. In all cases we assumed a linear coupling between the following and leading system. The fold-fold coupled system has been considered before in another
context (Brummitt et al., 2015), where it also has been noted that in systems with more than two coupled fold cascades not all subsystems undergo tipping ('hopping'). Moreover, stochastically coupled multi-stable systems have been considered in networks, where different types of domino effects can occur depending on the synchrony of the transition in the different network nodes (Ashwin et al., 2017; Creaser et al., 2018). Here we consider only two coupled systems, but allow different types of bifurcations, and the systems are physically coupled in a directional way.

We have discussed statistical indicators and analysis tools for cascading tipping points. Indicators for cascading tipping points are found in detrended cross correlation analysis (DCCA) and a special case of extrapolation using the DFA of the following system. These tools were applied in simulations involving both the double-fold and fold-Hopf cascades. The increased variance, AR(1) and DFA scaling exponent are clearly found in each case of single tipping. The cross-correlation indicators (DCCA and $\rho_{DCCA}$) did not evolve much throughout the time series, which indicates their insensitivity with respect to prox-
imity to single tipping points. Several limitations on the use of these variables have been mentioned. However, it seems that these metrics are highly sensitive to window and segment sizes, which leaves their potential as early warnings of cascading transition events inconclusive. The ratios of autoregressive metrics before and after the first transition seem to be a stronger warning of cascading transitions. More research is needed to exactly quantify these metrics.

The concept of cascading tipping was applied to study the behavior of a model describing a link between the Atlantic
Meridional Overturning Circulation (MOC) and ENSO. We modelled this using a coupling between the Stommel (1961) model and the Timmermann et al. (2003) ENSO-model by a meridional temperature gradient-dependent term in the external wind stress of the ENSO model. Through analysis of the bifurcation diagrams and simulations, a cascading tipping event is indeed possible in this case and our results are presented in the light of the fold-Hopf cascade.

A potential example of a double-fold cascade, that was not further treated here, could be the impact of a bistable MOC
on the (bistable) land ice formation on the Antarctic continent. In this case the coupling exists through the atmospheric $CO_2$ concentration, which depends on mixing and circulation in the ocean while strongly determining the existence of an ice sheet (DeConto and Pollard, 2003). During the Eocene-Oligocene transition, where a large ice sheet grew on Antarctica, a two-step two-step signal is observed in the deep-sea $\delta^{18}O$ ratio, suggesting two abrupt transitions. Using a box model by Gildor and



Tziperman (2000), Tigchelaar et al. (2011) showed that a two-step signal can be produced by first a MOC transition which changes the $CO_2$ concentration such that a transition occurs in the land-ice model.

These two applications indicate that there are likely many cases in which these cascading events occur in climate and therefore highlight the importance of the topic. Future research will point out whether these events are likely to happen in the future climate and whether these effects also occur in other fields than climate science. Of course, the theory covers the very basics of deterministic cascading events. One can imagine a wide range of phenomena if more complicated transitions between attractors are considered and when noise is included. For example, when a leading chaotic system is coupled to a deterministic following system with a back-to-back saddle-node bifurcation structure, a slight change in the chaotic attractor may change the background conditions for the following system such that it undergoes a transition. An application here may be the effect of a midlatitude atmospheric jet on the Atlantic MOC. We hope that this paper will stimulate more research on the various types of cascading tipping and also on the development of well-suited indicators and early warnings of such events.

*Acknowledgements.* AvdH and HAD acknowledge support by the Netherlands Earth System Science Centre (NESSC), financially supported by the Ministry of Education, Culture and Science (OCW), Grant no. 024.002.001. AvdH thanks Peter Ashwin for discussions in relation to this work and thanks the University of Exeter and the EPSRC funded Past Earth Network (Grant number EP/M008363/1) for funding an extended visit to the University of Exeter during the Summer of 2017.



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





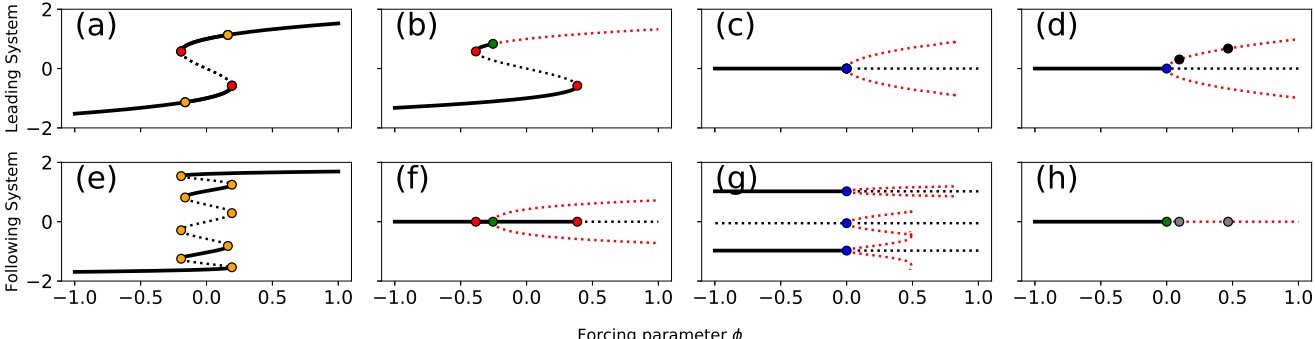

**Figure 1.** Bifurcation diagrams of coupled dynamical systems that show various cascading tipping types. Forcing parameter $\phi$ of the leading system is used on the horizontal axis, the leading system's (top) or following system's (bottom) equilibria are shown on the vertical axis. Lines indicate stable equilibria (black solid), unstable equilibria (black dashed) and oscillatory equilibria (red dashed lines indicate non-zero amplitudes in pairs). Dots indicate important bifurcation points: limit points (red/orange for the leading/following system), Hopf bifurcation points (blue/green for the leading/following system) and torus bifurcation points (black/grey for the leading/following system).

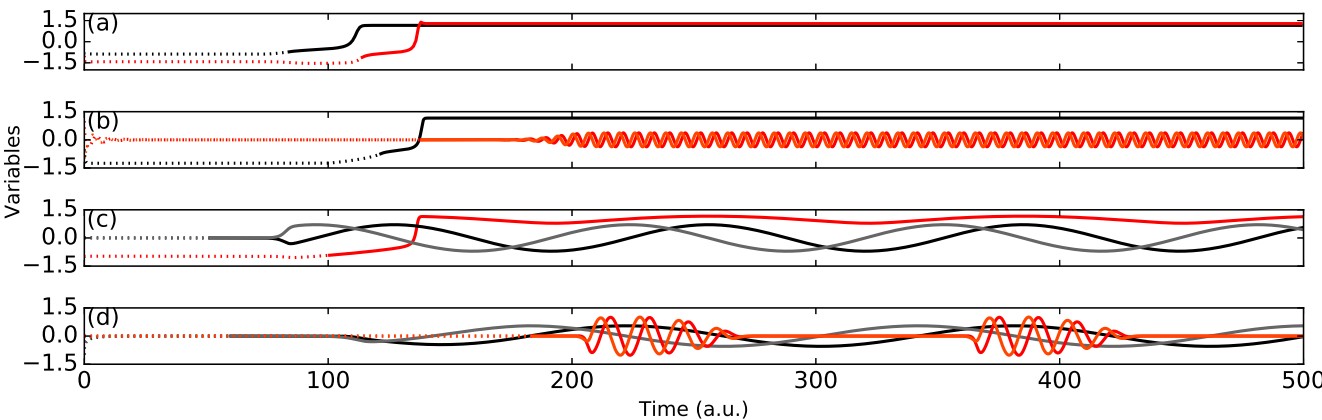

**Figure 2.** Example simulations for each cascading event type: the double-fold cascade (a), the fold-Hopf cascade (b), the Hopf-fold cascade (c) and the double Hopf cascade (d). Black and grey lines indicate the leading systems, red and orange lines indicate the following systems. Dotted lines indicate time before the critical threshold in the forcing $\phi(t)$ (black/grey) or coupling $\kappa(x)$ (red/orange) is reached, solid lines indicate the time after this. Parameter values for the modelled systems are given in Table 1.



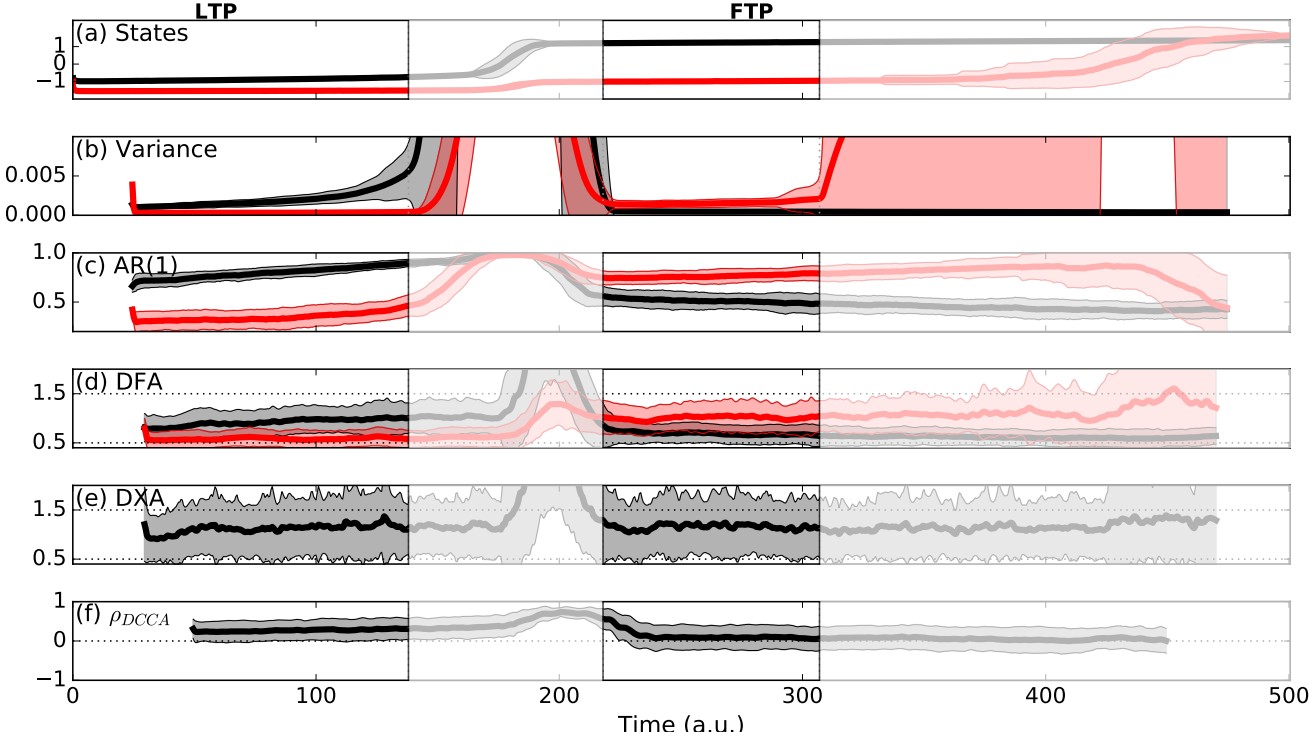

**Figure 3.** Ensemble (100-member) simulations of a dynamical system undergoing a double-fold cascade (Eqn. 13) where both systems undergo a transition, parameter values as in Table 2; (a) states of $x$ (black) and $y$ (red), (b) variance of $x$ (black) and $y$ (red), (c) autoregressive coefficient at lag 1 of $x$ (black) and $y$ (red), (d) detrended fluctuation analysis scaling exponent of $x$ (black) and $y$ (red), (e) detrended cross-correlation analysis scaling exponent and (f) detrended cross-correlation coefficient by Zebende (2011). White-shaded areas indicate windows containing the actual transitions. The increased variance, AR(1) and DFA scaling exponent prior to transition in the leading system and following system, respectively confirms the predicted increased memory through critical slowing down.





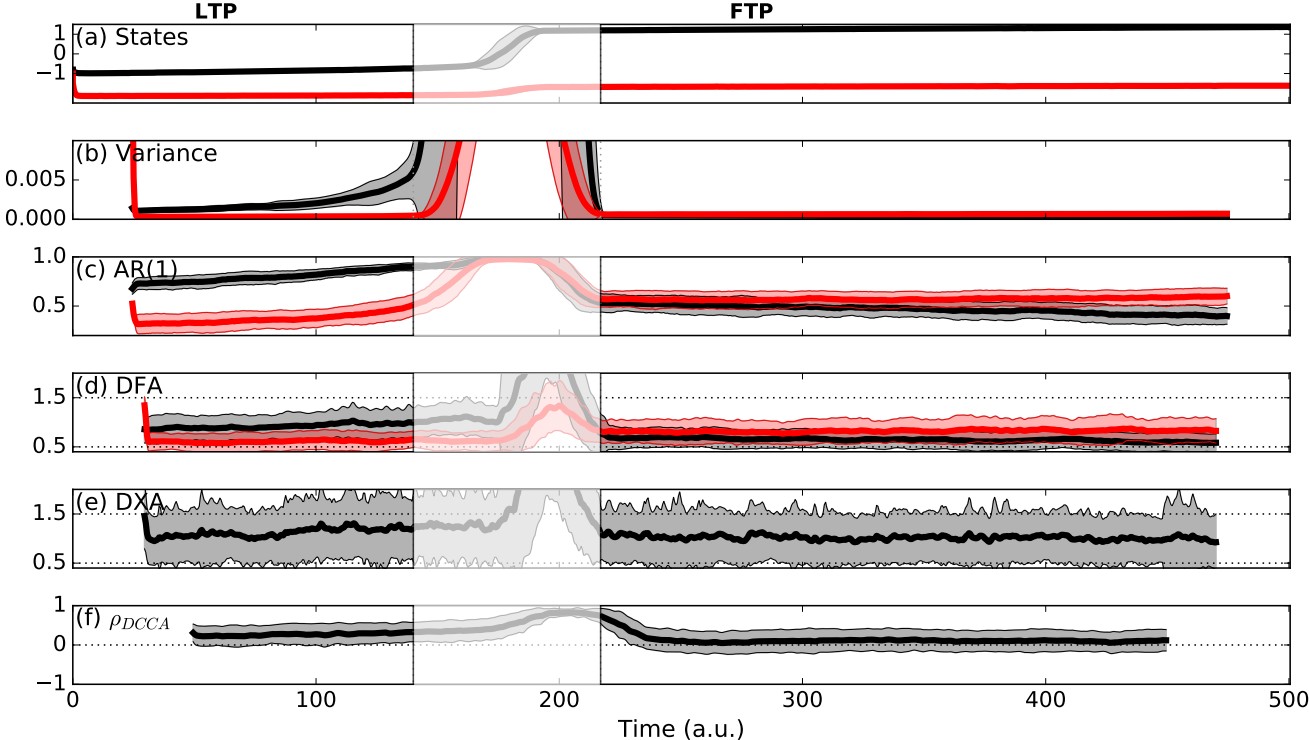

**Figure 4.** As in Fig. 3, but without any transition in the following system. In this case, only the leading system has a transition. Parameter values are given in Tab. 2.





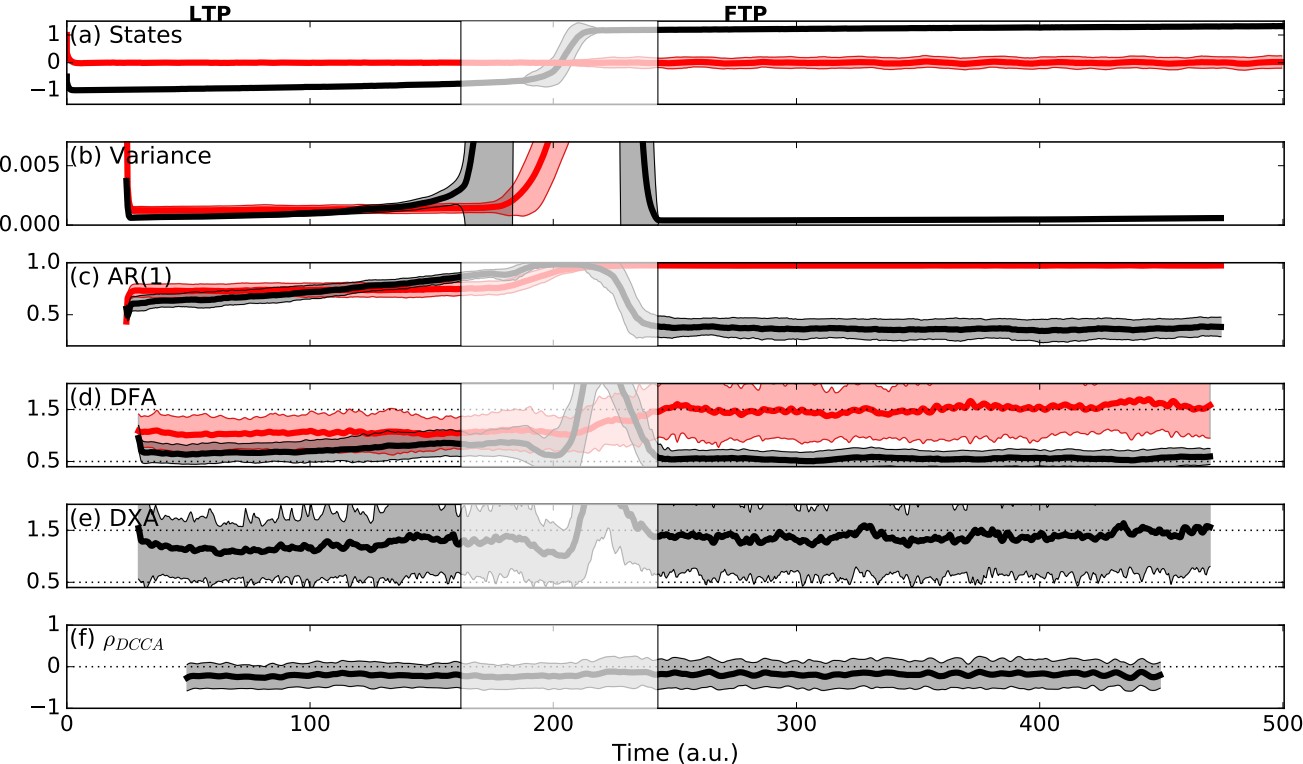

**Figure 5.** As in Fig. 3, but for the fold-Hopf cascade (Eqn. 14). Parameter values are given in Table 2.



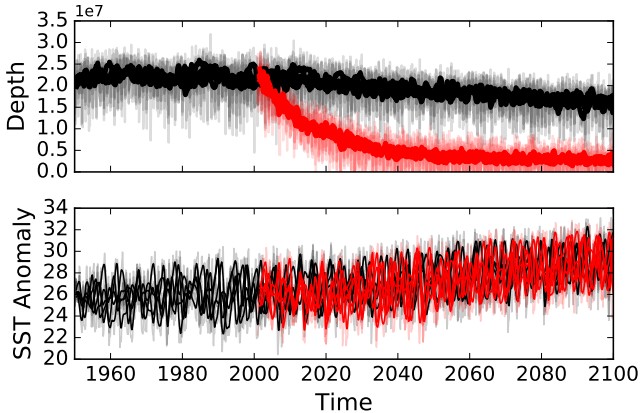

**Figure 6.** Top panel: Evolution of the five standard SRES-A1b runs (blue) and five HOSING-1 runs (red) in terms of the overturning. Bottom panel: NINO3.4 of the standard SRES-A1b ensemble (blue) and the HOSING-1 ensemble (red). Shaded thin lines indicate monthly means, thick lines indicate the deseasonalised values.

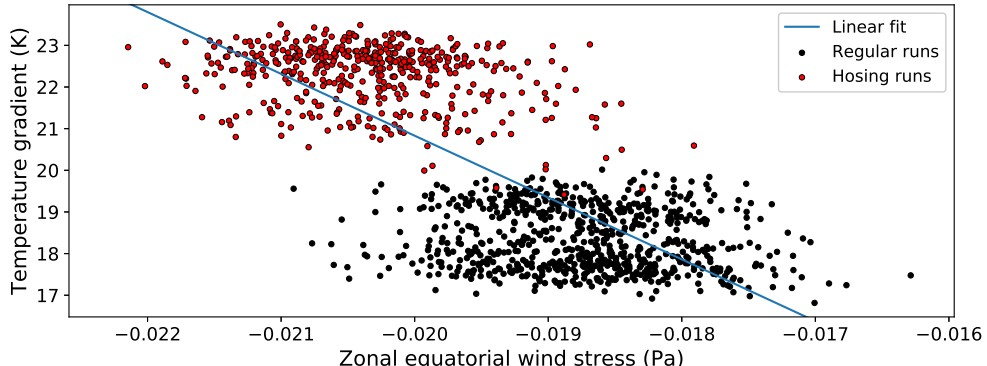

**Figure 7.** Zonal equatorial wind stress versus Atlantic temperature gradient. Data from the ESSENCE (Ensemble SimulationS of Extreme weather events under Nonlinear Climate changE) project is used, where the black dots refer to five members of the standard ensemble with SRES-A1b forcing (period 1950-2100), and the red dots refer to five members of the HOSING-1 ensemble where in 2000 a freshwater perturbation is applied (i.e., period 2000-2100). Five year running mean is applied, yearly averages are shown. The zonal equatorial wind stress here is defined as the average zonal wind stress over the latitudonal band 0-10°N. The Atlantic temperature gradient is defined as the difference between the SST in a northern box (50-60°N, 50-20°W) and a southern box (0-20°N, 45-20°W). The blue line indicates a linear fit.



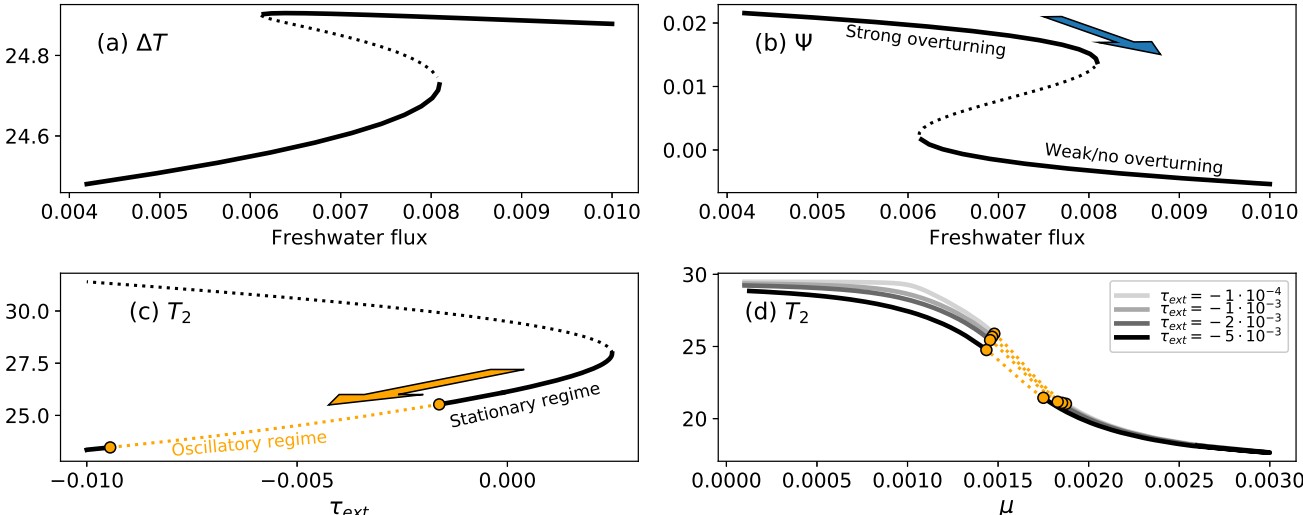

**Figure 8.** Bifurcation diagrams and forward runs of the Stommel (top panel) and of the Timmermann (bottom panel) models. Blue arrow indicates the collapse of the overturning circulation, which amplifies (negatively) the external zonal wind stress in the Pacific $\tau_{ext}$, such that (orange arrow), the system enters an oscillatory state. Orange arrow indicates subsequent tipping in the following (ENSO) system. Top panels: (a) Meridional temperature gradient equilibria versus freshwater flux, (b) non-dimensional stream function versus freshwater flux. These figures show the multiple states of the overturning. Bottom panels: (c) eastern equatorial Pacific SST versus $\tau_{ext}$ (for $\mu = 0.00145$), showing a regime where the system is stationary and a regime where the system is oscillatory, (d) eastern equatorial Pacific SST versus $\mu$ for different values of $\tau_{ext}$. Orange dots indicate Hopf bifurcation points, orange dotted lines indicate oscillatory regimes. Black and grey solid lines indicate stable equilibria, black dashed liens indicate unstable equilibria.

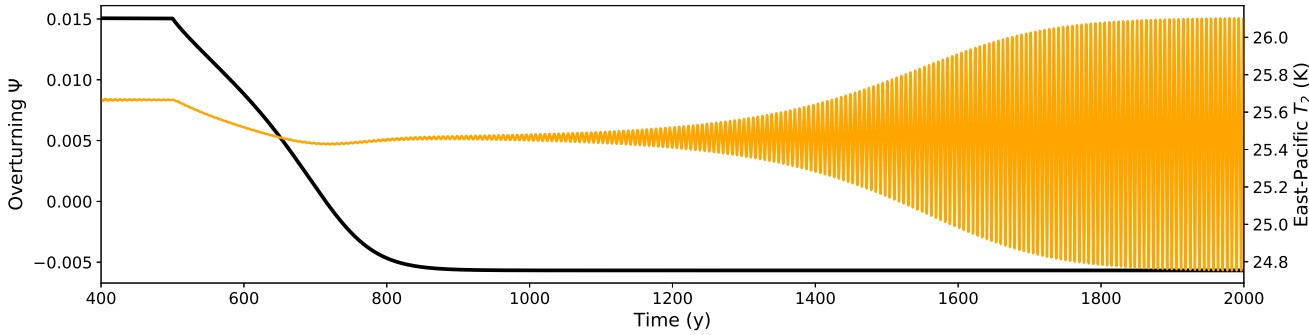

**Figure 9.** Simulation run of the coupled Stommel-Timmermann model for different model configurations, where the collapse of the overturning flow function (black) leads to the crossing of a Hopf bifurcation in the eastern equatorial-Pacific SST (orange).



**Table 1.** Parameter values and coupling for the four types of cascading tipping as shown in Figures 1 and 2.

| | **Double fold** (Eqn. 3) | **Fold-Hopf** (Eqn. 4) | **Hopf-fold** (Eqn. 5) | **Double Hopf** (Eqn. 6) |
|---|---|---|---|---|
| **Leading system** | | | | |
| | $\phi_c = \pm 0.19$ (Fold) | $\phi_c = \pm 0.38$ (Fold) | $\phi_c = 0$ (Hopf) | $\phi_c = 0$ (Hopf) |
| | Bistable for | Bistable for | Oscillatory for | Oscillatory for |
| | $\lvert \phi \rvert < \sqrt{\frac{-4a_1^3 a_2^3}{27 a_1^4}}$ | $\lvert \phi \rvert < \sqrt{\frac{-4a_1^3 a_2^3}{27 a_1^4}}$ | $\phi > 0$ | $\phi > 0$ |
| | (if $a_1 < 0, a_2 > 0$) | (if $a_1 < 0, a_2 > 0$) | (if $a_1 b_1 < 0$) | (if $a_1 b_1 < 0$) |
| **Coupling** | | | | |
| | $\gamma = 0.48x$ | $\gamma = -0.1 + 0.12x$ | $\gamma = 0.05 + 0.5x$ | $\gamma = -0.05 + 2x$ |
| **Following system** | | | | |
| | $\gamma_c = \pm 0.54$ (Fold) | $\gamma_c = 0$ (Hopf) | $\gamma_c = \pm 0.38$ (Fold) | $\gamma_c = 0$ (Hopf) |
| | Bistable for | Oscillatory for | Bistable for | Oscillatory for |
| | $\lvert \gamma \rvert < \sqrt{\frac{-4b_1^3 b_2^3}{27 b_1^4}}$ | $\gamma > 0$ | $\lvert \gamma \rvert < \sqrt{\frac{-4c_1^3 c_2^3}{27 c_1^4}}$ | $\gamma > 0$ |
| | (if $b_1 < 0, b_2 > 0$) | (if $b_1 c_1 < 0$) | (if $c_1 < 0, c_2 > 0$) | (if $c_1 d_1 < 0$) |
| **Parameters** | | | | |
| | $a_1 = -0.5$ | $a_1 = -1$ | $a_1 = 0.05; a_2 = 1$ | $a_1 = 0.04; a_2 = 2$ |
| | $a_2 = 0.5$ | $a_2 = 1$ | $b_1 = -0.05; b_2 = 1$ | $b_1 = -0.04; b_2 = 2$ |
| | $b_1 = -0.5$ | $b_1 = b_2 = 1$ | $c_1 = -1$ | $c_1 = 0.4; c_2 = 1$ |
| | $b_2 = 1.0$ | $c_1 = -1; c_2 = 1$ | $c_2 = 1$ | $d_1 = -0.4; d_2 = 1$ |



**Table 2.** Parameter values, coupling and initial conditions for the ensemble simulations of the Double Fold and Fold-Hopf systems as shown in Figures 3, 4 and 5.

| **Double fold** (Eqn. 13) (following system tips) | **Double fold** (Eqn. 13) (following system does not tip) | **Fold-Hopf** (Eqn. 14) |
|---|---|---|
| **Forcing and coupling** | | |
| $\phi(t) = 0.0012t$ | $\phi(t) = 0.0012t$ | $\phi(t) = 0.002t$ |
| $\gamma(x) = 0.05 + 0.37x$ | $\gamma(x) = 0.05 + 0.37x$ | $\gamma(x) = -0.2 + 0.3x$ |
| **Parameters** | | |
| $a_1 = -0.5$ | $a_1 = -0.5$ | $a_1 = -1$ |
| $a_2 = 0.5$ | $a_2 = 0.5$ | $a_2 = 1$ |
| $b_1 = -0.5$ | $b_1 = -0.25$ | $b_1 = 0.1; b_2 = 1$ |
| $b_2 = 1.0$ | $b_2 = 1$ | $c_1 = -0.5; c_2 = 1$ |
| **Integration time** | | |
| $t_{max} = 500$ | $t_{max} = 500$ | $t_{max} = 500$ |
| $\Delta T = 0.5$ | $\Delta T = 0.5$ | $\Delta T = 0.5$ |
| **Noise** | | |
| Noise mean = 0 | Noise mean = 0 | Noise mean = 0 |
| Noise variance = 0.1 | Noise variance = 0.1 | Noise variance = 0.1 |
| **Initial conditions** | | |
| $(x_0, y_0) = (-0.8, -1)$ | $(x_0, y_0) = (-0.8, -1)$ | $(x_0, y_0, z_0) = (-0.5, 1, -1)$ |





**Table 3.** Comparison of the ratios of autoregressive variables prior to and after the first transition, using the ensembles shown in Fig. 3 and Fig. 4.

| Ratios | Mean | Standard deviation |
|---|---|---|
| **With second tipping** | | |
| | | |
| Leading variance | 0.24 | 0.44 |
| Leading AR(1) | 0.62 | 0.09 |
| Leading DFA | 0.79 | 0.27 |
| Following variance | 3.95 | 1.53 |
| Following AR(1) | 1.92 | 0.39 |
| Following DFA | 1.70 | 0.49 |
| **Without second tipping** | | |
| | | |
| Leading variance | 0.15 | 0.06 |
| Leading AR(1) | 0.60 | 0.08 |
| Leading DFA | 0.74 | 0.23 |
| Following variance | 1.63 | 0.42 |
| Following AR(1) | 1.34 | 0.31 |
| Following DFA | 1.40 | 0.41 |

**Table 4.** Results of Student's t-test on the differences between the ratios (in Tab. 3) of the cases *with* and *without* second tipping. *p-values calculated using the scipy.stats Python package.

| t-test variable | T-statistic | dF | p-value* |
|---|---|---|---|
| Leading variance | 1.62 | 101.67 | 0.11027 |
| Leading AR | 1.32 | 197.07 | 0.18904 |
| Leading DFA | 1.23 | 190.60 | 0.22264 |
| Following variance | 13.73 | 112.38 | 0.0 |
| Following AR | 9.93 | 184.12 | 0.0 |
| Following DFA | 4.13 | 190.17 | 0.00006 |



**Table 5.** NINO3.4 statistics (of deseasonalised data) for the different ensembles. The uncertainty stated is the standard deviation among the five runs within the ensemble. It is visible that in the case of a collapsed overturning, El-Niño intensifies more than without a collapsed overturning.

| Time period | Ensemble | Variable | Value |
|---|---|---|---|
| 1950-2000 | Standard SRES-A1b | Mean | $25.86 \pm 0.046$ |
| | Standard SRES-A1b | Variance | **$1.705 \pm 0.447$** |
| 2001-2100 | Standard SRES-A1b | Mean | $27.51 \pm 0.032$ |
| | Standard SRES-A1b | Variance | **$2.581 \pm 0.112$** |
| 2001-2100 | HOSING-1 | Mean | $27.27 \pm 0.053$ |
| | HOSING-1 | Variance | **$3.21 \pm 0.42$** |