# Peer review of "Cascading transitions in the climate system"

_Earth System Dynamics, 2018_

## Referee Comment (RC1) · A. Tantet (Referee) · 2 Jun 2018

The authors set up a mathematical framework for the study of cascading tipping points based on the extension of bifurcation theory and normal forms to coupled critical systems. After a numerical study of the potential early-warning indicators for cascading transition, the authors apply this framework to the interaction of the AMOC collapse and ENSO strengthening observed in a hosing experiment with a large ensemble of simulations from a General Circulation Model (GCM). For that purpose, they use a minimal coupled model for which the bifurcation diagrams can be calculated. The aim of this study is not to set up a complete mathematical theory for the study of cascading bifurcations, but rather to bring theoretical support to study successive abrupt climate changes, such as observed in paleoclimatological records during the Eocine-Oligocene transition. Application to climate variability is particularly relevant, since abrupt climate changes have happened in the past. Moreover, abrupt changes are found in a hi-

erarchy of climate models, although it is not yet clear whether GCMs used for the Intergovernmental Panel on Climate Change assessment reports are able to resolve them properly. The application to the AMOC collapse and ENSO is particularly relevant, since the melting of Greenland as a result of global warming is expected to lead to the collapse, or at least a weakening of the AMOC. In addition, the article is clear, well written and fit well the scope of Earth System Dynamics. I therefore recommend it for publication after minor revisions.

Specific comments:

Section 2:

- The authors first describe possible scenarios of cascading tipping by combining the normal forms most relevant for applications and involving only one or a pair of stability exponents crossing the imaginary axis. As such, the framework is suited for coupled systems for which both the leading and the following systems are close to a saddle and/or a Hopf bifurcation, a situation relevant for the applications considered here. However, the climate system is a high-dimensional system with a large number of positive Lyapunov exponents, whereas the bifurcations considered here involve only one or two-dimensional attractors rather that chaotic sets. As such, while the mathematical framework considered here appears to be an important direction to explore for climate applications, I would consider it only as a first important step towards understanding more complex abrupt climate changes, such as the one studied in section 4. This point could be discussed more by the authors.

- In bifurcations involving meta-stable states, such as the double saddle node bifurcation, or bifurcations involving strange attractors (e.g. (Tantet, Lucarini, Lunkeit, & Dijkstra, 2018)âĄă), a critical transition occurs through a saddle point, or a strange saddle. In this case, although the saddle set is globally unstable, its stable manifold may be responsible for a slowing down at the vicinity of the saddle, resulting in what also looks like a two step transition. Could you discuss why the cascading bifurcations may or may not be a better candidate to explain the two-steps transitions such as observed during the Eocene-Olegocene transition?

Section 4:

- In Fig. 7, there is indeed a strong correlation between the temperature gradient and the wind stress. However, as the author remark, there is also a strong spread, which should result in a strong variance in the estimate of the coefficients in Eq. 21. Could you use an ensemble method such as bootstrapping or a Bayesian model to test the probability that such a cascading tipping indeed occurs when sampling the different values of the coefficients of Eq. 21? This would allow to discuss the robustness of the results to the dependence of the wind stress on the temperature gradient.

- You explain well how the parameters of the wind stress equation are estimated from the model runs. However, it is not clear to me how the parameters of the Stommel and of the Timmermann are chosen. Are the parameter values used the same as in the references? Are they chosen so as to be as close as possible to historical data? So has to reproduce the mean state and variability found in observations? Or so as to favor the occurrence of the cascading tipping? In any case, I understand that estimating the parameter values of minimal models from observations or complex models is a difficult and not always relevant task. However, the sensitivity of the occurrence of the cascading tipping on the parameters of the coupled Stommel-Timmermann model should be discussed to better assess the likelihood of such tipping to occur.

Discussion:

- Salinity biases, such as found in the GCM used in this study, have shown to have a strong impact on the bi-stability of the AMOC (Mecking, Drijfhout, Jackson, & Andrews, 2017). Considering that the strengthening of ENSO also occurs in the control run, could you discuss whether this is/is not an important factor to take into account when asking whether or not such a cascading tipping of the AMOC+ENSO system could occur in the future.
References:

Mecking, J. V., Drijfhout, S. S., Jackson, L. C., & Andrews, M. B. (2017). The effect of model bias on Atlantic freshwater transport and implications for AMOC bistability. Tellus, Series A: Dynamic Meteorology and Oceanography, 69(1), 1–15. http://doi.org/10.1080/16000870.2017.1299910 Tantet, A., Lucarini, V., Lunkeit, F., & Dijkstra, H. A. (2018). Crisis of the Chaotic Attractor of a Climate Model: A Transfer Operator Approach. Nonlinearity, 31(5), 2221. http://doi.org/https://doi.org/10.1088/1361-6544/aaaf42 âĄă

———————————————

---

## Referee Comment (RC2) · Anonymous Referee #2 · 2 Jun 2018

The article "Cascading transitions in the climate system" by M. Dekker et al. conceptually explores "cascades" of tipping points. The phenomenon is interesting and relevant in many contexts, and the authors focus on the climate system here and present a modelling example. Tipping points and their precursors are a popular topic in many articles but to my knowledge they have not been analysed in this specific context before. The technical quality and presentation of the article (structure, language and figures) are on a relatively high level. To this extent, the manuscript merits publication in Earth System Dynamics.

There are however also some fundamental questions that I think are important to be (re-)assessed and clarified, and the manuscript should be revised accordingly:

1. It is not very clear to me what the authors see as the main aim of the paper. In the abstract it is stated that they aim at providing a new theory / a mathematical frame-

work. I am not convinced that these specific claims are supported by the contents of the paper. For example, isn't a theory something that provides an explanation for a certain number of facts? What is the new explanation here, and of what? I like how Sect. 2 systematically explores conceptual models of two combinations of generic bifurcations. In this section, I have the impression that one tipping point immediately triggers the next (instead of the tipping point in the leading system only bringing the following system closer to its own tipping point, which is then again triggered by the changing control parameter)? If this is the case, how can early warning signals even be used to predict the second transition? I will elaborate on this point below. In general, the early-warnings analysis in Sect. 3 is less clear to me than Sect. 2. I have the impression that the authors present two analyses of a single tipping, and not one analysis of an induced tipping, which somewhat questions the novelty of the approach. At first, I thought that the authors aim to predict the second tipping before the first, or infer what kind of bifurcation to expect. However, after the first examples of cascading tipping it seemed like the approach was to use early warning signals to first predict the first transition and after that predict the second transition, but to do that the concept of cascading tipping is not necessary, since they basically predict two tipping points independent from each other. In this context, I was also wondering why the external shock that the second system receives must result from a bifurcation in the leading system. Could it not also result from other kinds of tipping points, or a sudden step or peak in forcing like a volcanic eruption or a pulse release of greenhouse gases? Why is the leading system needed at all when the main aim is to detect if the first shock will trigger a transition in the following system? The model example in the end (ENSO-AMOC model) is interesting, but its purpose is not clear enough to me. Maybe the authors can clarify what it is that they want to demonstrate exactly and state this clearly in the introduction and draw conclusions using the results they show. The conclusion section should be extended by a discussion about what questions are answered and what the implications of the results are. What can we do or understand with the approach in this paper that we were not able to do or understand before? What should be done next?

2. The reasons for the choice of methods should be explained better. This is often linked to the problem mentioned above, i.e. the lack of clarity about the aim of the study. Once this aim becomes clearer, it should also be easier to explain why certain methods are applied.

2.1. Specifically, the choice of statistical indicators needs better justification. I currently do not see what the early warnings approach can add to previous studies. For example, why is DFA used as a warning signal instead of just the autocorrelation? Since autocorrelation is simpler to calculate and more intuitive, I would like to see an argument for the added value of DFA. The statement that "standard quantities not always provide an early warning signal" (page 6, line 26/27) should be backed up with an argument and references, and then it should be explained why DFA can cope with this. I would actually expect DFA to fail whenever autocorrelation fails, which happens when the system is more complicated than the typical Langevin equation / AR1-process with one fixed time scale. One argument the authors give is that "DFA copes well with non-stationarity". First: What is the explanation for this statement? What is the tradeoff when using DFA (more data needed?). Second: Couldn't one just remove non-stationarity with a high-pass filter (which is what the authors seem to do already) and then use traditional early warning signals? The authors use relatively simple models here, where the parameter can be varied as slowly as necessary to remove non-stationarity (or they could even make long stationary time series for different fixed parameter values). Another argument the authors provide is that DFA captures long-range correlations. But why should one expect such long-range correlations in the simple models the authors use? Can they even exist? So, in a nutshell, why is DFA needed in this paper? Then the authors generalise DFA to capture the involvement of several state variables (using DCCA). This could make sense if they were trying to detect something about the coupled system, for instance, which variable is leading, what will happen after tipping 1 and 2. However, the main results seem to consist in predicting tipping 1, and then detecting that the following system has moved closer to a tipping point (by the way, how do we know that there is a second tipping point? The

fluctuations could just have changed for another reason.). As far as I can see, DCCA is not needed to do so, an AR1-analysis of each single variable may have sufficed. The explanation on page 7, lines 26-30 is unclear to me. In what way and to what purpose and why can Pearson's correlation "not be used"? And what is meant with a "one-to-one-relationship" (line 30)? Sect. 3.3.1: The authors state several times that DXA and DCCA are sensitive to the segment size and moving window size, but have different values been tried? It would be nice to show how sensitive they are, and what this means for the results. It could help already to just show more runs with different parameter settings. In section 3.1 The essential part of degenerate fingerprinting is the projection on the leading EOF in a multivariate system. However, the manuscript skips this part of the method, and therefore, right now, just explains the lag-1 autocorrelation and not degenerate fingerprinting. Could one learn something about a system with cascading tipping points by using degenerate fingerprinting on the multivariate signal? Sect. 3.3: Why are only the double-fold and fold-Hopf systems tested for the early-warning approach, and not the two systems with a Hopf bifurcation in the leading system? This choice should be explained or the other two examples should be included as well. Sect. 3.3.2, page 10, last paragraph: The oscillation seems to affect the measurement of autocorrelation. I think that one should here measure the autocorrelation of the residuals around a mean oscillation, either by subtracting this mean cycle somehow, or by defining a period and working with Poincare sections (snapshots after each period). Otherwise the result would probably be meaningless.

2.2 In both figure 1 and figure 2, the choice for the coupling of the two subsystems seems to be arbitrary. These choices could be explained better to make it more understandable for the reader. For example, one can shift the two systems versus each other (by varying parameter gamma1), such that the two tippings are well separated, or that they are really intertwined (one tipping inducing the other immediately). How would the stability landscape then look like, and what would we see in early-warning signals? I was also wondering why the values of gamma1 have been chosen in a way that gamma1 is 0 for the double-fold, <0 for the Fold-Hopf, and double-Hopf, and

>0 for the Hopf-fold. Conceptually it would make a difference if the second tipping is triggered by the changing parameter or a direct consequence of the first tipping. It seems that the latter is always the case here for all parameter choices? This should be made clear from the beginning (as I mentioned above). Probably related to this point: In figure 2 it seems to be the case that the leading system tips before the following system, whereas the bifurcation plots seem to indicate that this happens at the same time. Where does this time delay come from? Similarly, a time delay can be seen in the Fold-Hopf system (Fig. 2b), while Fig. 1b would make me expect a discontinuous jump from a stationary solution to a cycle with some non-zero amplitude. Also, according to Table 2, the control parameter Phi increases linearly with time, but I do not see any change in state (or the amplitude of its oscillations) in Fig. 2, and on page 5, last paragraph, it is mentioned that at some point the amplitude would jump to a large value when both equilibria are accessed, but this is not seen in the Figure. In this context, Fig. 1 and 2 appear contradictory to me. This point is actually a crucial one because the period between the two tippings is used to detect early warning signals for the second tipping. How can it even be that there is enough time to detect them, when the system is already in the process of tipping? Here it looks like the second tipping is actually not caused directly by the first, but by the changing control parameter (in contrast to the impression I got in the previous section). If it is a real cascade (tipping 2 directly induced by tipping 1), wouldn't the system's state suddenly be very far from equilibrium after tipping 1. Can early warnings even be expected in this situation (mind they sample the equilibrium when the state fluctuates around it)? Moreover, I imagine that the relative time scale between the systems matters (controlled by the different coefficients in the equations). For example, in case of the Hopf-fold system, it would matter how fast and how large the oscillation in the leading system is compared to the following system's response time. So why has this particular coupling been chosen for the paper, and how representative is that for the climate system?

2.3 - The climate model (coupled ENSO and AMOC) seems very interesting. However, it is not completely clear to me what point exactly the authors want to make by showing

it. Sect. 4.3 is very short and I don't really understand its purpose. In the conclusion section and in the abstract it seems to be argued thatit illustrates that cascading tipping can occur in climate models, but as this is already known according to the introduction, and given that the model has been designed like this on purpose, what new information does this model provide? - Also, it should be more clearly explained how the two existing models have been coupled. I found it difficult at first to identify the common variables in the models that were linked. More precise wording might help (e.g. "through influence of the wind stress" - influence on what?, "in the original model" - which model?). It seems that the authors introduce an equation for the wind stress tau which links tau from the ENSO model to the temperatures from Stommel's model? Then one could say so from the start, followed by the details. - The model seems to be a representation of the Fold-Hopf case above? This should be explicitly stated from the beginning. - Why have the authors not done an early-warning analysis with this AMOC-ENSO model? This would be a natural step to do after the generic models above. The authors use data from a complex model to tune their conceptual model. What can we learn from that data directly about predicting each tipping, or the coupling (or whatever the authors aim to do)? Could one apply a statistical analysis and infer something about that model from the data?

Minor comments

- What I find most interesting is the analysis of the coupled deterministic systems, e.g. in Fig. 1. A very interesting aspect is the occurrence of intermediate (in terms of the state variable) stable states which are inaccessible when varying the control parameter. It seems that only noise can bring the system on these branches. This aspect is however not discussed in the paper. It is of course up to the authors if they want to go into this, but I would recommend them to at least comment on these hidden states, which I personally find more novel and exciting than the early-warning part of the paper. Could there be such hidden stable states in the climate system and how can they be found?

- Section 2.1 + Figure 1: It took me quite some time to understand what is going on. It could be helpful to create an X-Y bifurcation plot in addition to the phi-X plots and phi-Y plots that are shown already (to see how each system behaves in isolation). More emphasis can be put on explaining this figure, because this in itself is already an interesting result. The authors might even think of making an animation as extra material, to show how the subplots relate to each other. Also, it could be nice to show how figure 2 relates to figure 1.

- Several different names are sometimes used for the same thing, at least for the fold bifurcation (fold / back-to-back / back-to-back saddle-node). I had never come across the term back-to-back before. Is one term a subset of another? The authors should clarify this and unify the language.

- Some of the references are a bit outdated (e.g. Kutzbach 1996 on page 2; a lot has happened since then), or could be a bit more specific. Page 2: Scheffer 2009 is a review of some of the earlier papers like Held 2004, some of which are cited later; Peng 1994 is not about predicting tipping points. Also, note that there are papers from the 80ies dealing with statistical precursors already, e.g. by Wiesenfeld, 1984. page 1 (lines 17ff): Lenton et al. 2008 do not show evidence that there are tipping points in the climate system (though the paper is often cited in that way), so this paragraph should be formulated more cautiously. Also, the vegetation states found by Hirota et al. are purely ecological phenomena, and do not imply any tipping points in the climate.

- page 6, line 18: "close to critical transition" (2x), should be "close to a critical transition".

- page 10, line 17/18: "as it is no critical transition": why not? And what is a critical transition?

- In Fig. 8, I found it confusing that the labels are not next to the vertical axes but inside the figure. I do understand that this is consistent with the previous figures, so I don't have strong feelings about this.

---

## Author Comment (AC1) · 23 Jul 2018

*We thank the referee for the careful reading and the useful comments and will adapt the manuscript accordingly. Below is a point by point reply with the referee's comments in bold font, our reply in italic font and the changes in manuscript in normal font.*

1. Comment of the referee:

**Section 2: The authors first describe possible scenarios of cascading tipping by combining the normal forms most relevant for applications and involving only one or a pair of stability exponents crossing the imaginary axis. As such, the framework is suited for coupled systems for which both the leading and the following systems are close to a saddle and/or a Hopf bifurcation, a situation relevant for the applications considered here. However, the climate system is a**

[Figure]

**high-dimensional system with a large number of positive Lyapunov exponents, whereas the bifurcations considered here involve only one or two-dimensional attractors rather that chaotic sets. As such, while the mathematical framework considered here appears to be an important direction to explore for climate applications, I would consider it only as a first important step towards understanding more complex abrupt climate changes, such as the one studied in section 4. This point could be discussed more by the authors.**

Author's reply:
*We agree that abrupt climate changes in reality are connected to more complex chaotic sets and impossible to attribute to a single bifurcation or two bifurcations. As the referee also points to, the aim of this paper is to give a framework of cascading transitions with mathematical examples, analyses and applications to conceptual models. The step towards the real climate system should be taken with care. Especially in the beginning of the paper this can indeed be made more clear. In the discussion section, this was already mentioned (e.g. page 16, line 5-7).*

Changes in text:
We will address the connection between the idealized cases of cascading tipping here and transitions in the real system in the revised introduction en discussion.

In the beginning of section 2 we will add: 'In this section, we present a mathematical framework for simple cascading transitions, that acts as a first step towards analysing the more complex transitions happening in reality.'

2. Comment of the referee:
**In bifurcations involving meta-stable states, such as the double saddle node bifurcation, or bifurcations involving strange attractors (e.g. (Tantet, Lucarini,**
**Lunkeit, & Dijkstra, 2018)), a critical transition occurs through a saddle point, or a strange saddle. In this case, although the saddle set is globally unstable, its stable manifold may be responsible for a slowing down at the vicinity of the saddle, resulting in what also looks like a two step transition. Could you discuss why the cascading bifurcations may or may not be a better candidate to explain the two-steps transitions such as observed during the Eocene-Oligocene transition?**

Author's reply:
*In DeConto and Pollard (2003), it is suggested that the atmospheric CO2 concentration influences the existence of an ice sheet on Antarctica, via its effect on the ice-albedo and height-mass balance feedbacks. As the box model by Gildor and Tziperman (2000) contains these feedbacks, and a boxed ocean in which Tigchelaar et al. (2011) found multiple steady states for the meridional overturning circulation, a cascading event (of two bistable systems) could be simulated here (as written on p. 15 line 29 to p. 16 line 2). Of course, the comparison with the Eocene-Oligocene transition as found in proxy records should be made with care, because of the simplicity of the model used in Tigchelaar et al. (2011). In the present manuscript we have added this mainly as an example, but clearly further work is necessary to substantiate the hypothesis of cascading tipping being relevant for the Eocene-Oligocene transition. In particular, the coupling of the two bistable systems via the carbon cycle (determining the atmospheric CO2) requires more attention. This goes beyond the scope of the present manuscript and will be elaborated on in a follow-up study.*

Changes in text:
We will cite and shortly discuss the Tantet et al. (2018) paper. In the revised discussion, we will add: 'Although from a physical perspective, this is a potential example of a cascading transition, we make no claim about whether such a transition likely occurred at the Eocene-Oligocene transition.'

3. Comment of the referee:
**Section 4: In Fig. 7, there is indeed a strong correlation between the temperature gradient and the wind stress. However, as the author remark, there is also a strong spread, which should result in a strong variance in the estimate of the coefficients in Eq. 21. Could you use an ensemble method such as bootstrapping or a Bayesian model to test the probability that such a cascading tipping indeed occurs when sampling the different values of the coefficients of Eq. 21? This would allow to discuss the robustness of the results to the dependence of the wind stress on the temperature gradient.**

Author's reply:
*This is a good suggestion. Note that the results shown in Fig. 8 and 9 are dependent on multiple parameters and choices made (not only the ones that are derived from Fig. 7). To be precise, these are the definition of the North Atlantic and the Equatorial Atlantic regions, the zonal wind stress region, the reference wind stress parameter ($\tau_0$), and it also turns out that the temporal resolution and running mean may dramatically change the values of $\alpha_\tau$ and $\gamma_\tau$ in Eqn. (21).*

Changes in text:
We will add such results (using bootstrapping with different values of $\alpha_\tau$ and $\gamma_\tau$ in Eqn. (21)) to the revised paper.

4. Comment of the referee:
**You explain well how the parameters of the wind stress equation are estimated from the model runs. However, it is not clear to me how the parameters of the Stommel and of the Timmermann are chosen. Are the parameter values used**

[Figure]

the same as in the references? Are they chosen so as to be as close as possible to historical data? So has to reproduce the mean state and variability found in observations? Or so as to favor the occurrence of the cascading tipping? In any case, I understand that estimating the parameter values of minimal models from observations or complex models is a difficult and not always relevant task. However, the sensitivity of the occurrence of the cascading tipping on the parameters of the coupled Stommel-Timmermann model should be discussed to better assess the likelihood of such tipping to occur.

Author's reply:
*For the Stommel and Timmermann models, we have used the standard values as in the original references, except when stated otherwise (for example in the case of the freshwater forcing in Eqn. 23). The parameter mu (that partly determines the closeness of the ENSO system w.r.t. the Hopf bifurcation) has been chosen to be near critical for the Timmermann model.*

Changes in text: We will explicitly mention how the value of mu (in caption of Fig. 9) was determined. On page 14 we will add: 'In the Stommel and Timmermann models, we use the standard parameter settings, as given in the references, unless stated otherwise.'

5. Comment of the referee:
**Discussion: Salinity biases, such as found in the GCM used in this study, have shown to have a strong impact on the bi-stability of the AMOC (Mecking, Drijfhout, Jackson, & Andrews, 2017). Considering that the strengthening of ENSO also occurs in the control run, could you discuss whether this is/is not an important factor to take into account when asking whether or not such a cascading tipping of the AMOC+ENSO system could occur in the future.**

Authors reply:

*Whether a cascading tipping event is what actually occurred in the HOSING-1 runs, is not known. Probably there is a more complicated reason behind the increased SST variance in HOSING-1 with respect to the standard runs, and likely a mix of different effects. The effect of salinity biases on the bimodality and hence on the AMOC-ENSO coupling is interesting but outside the scope of this paper. In the coupled Stommel-Timmermann model, we know in which parameter regime of the freshwater flux there is bimodality in the AMOC because that follows directly from the Stommel model's design.*

Changes in text:

In the revised discussion, we will shortly mention the effect of salinity biases on the bimodal behavior of the AMOC in GCMs, and its potential effect on the cascading behavior (and cite the relevant papers).

---

## Author Comment (AC2) · 23 Jul 2018

*We thank the referee for the careful reading and the useful comments and will adapt the manuscript accordingly. Below is a point by point reply with the referee's comments in bold font, our reply in italic font and the changes in the manuscript in normal font. We have taken the liberty to divide the comments into smaller pieces and discuss them separately.*

1. Comment of the referee:
**It is not very clear to me what the authors see as the main aim of the paper. In the abstract it is stated that they aim at providing a new theory / a mathematical framework. I am not convinced that these specific claims are supported by the contents of the paper. For example, isn't a theory something that provides an**

[Figure]

**explanation for a certain number of facts? What is the new explanation here, and of what?**

Authors reply:
*The aim of the paper is introducing the concept of cascading transitions. The contents of the paper support this by giving mathematical examples of these events, describing their statistics, and giving an example from climate physics. We agree that it is only a framework, not a complete theory (which was never the aim). In that respect, the first sentence of the abstract and the first sentence of Section 4 need to be rephrased, as they indeed imply that we provide a new theory.*

Changes in text:
Abstract, we will rewrite the first sentence to 'We introduce a framework of. . .' Section 4, in the first sentence we will change 'theory' to 'concept'.

2. Comment of the referee:
**I like how Sect. 2 systematically explores conceptual models of two combinations of generic bifurcations. In this section, I have the impression that one tipping point immediately triggers the next (instead of the tipping point in the leading system only bringing the following system closer to its own tipping point, which is then again triggered by the changing control parameter)?**

Authors reply:
*We understand the confusion and will try to make it more clear in the text. First of all note that the bifurcation diagrams only show equilibrium states of purely deterministic systems. This means that if the bifurcation diagram does not show that the following system tips after the leading system is forced with phi, then the following system will*

*never tip (there is no noise). So actually, one would expect that the bifurcations of the leading and following system overlap (when only looking in phi space), but not in time. The issue why the forward runs in Fig. 2 do show a gap between the two tipping points can be understood as follows. This is purely the transient as it takes a little time for the dynamical system to adapt to the new parameter setting (how long this takes may also depend on the specific form of the coupling). Fig. 2 nicely shows that the two tipping points are different events. The dashed lines show when the system is still stable in their old equilibrium, and the solid lines show whenever they are drawn to the second equilibrium. For example, in Fig. 2a one can see that the following system (red) only reaches its critical threshold to be drawn to the next equilibrium exactly when the leading system tips (so not before due to the increased forcing, but it is caused by the first tipping).*

Changes in text:
We will provide more explanation on the results of Fig. 2 in the text.

3. Comment of the referee:
**If this is the case, how can early warning signals even be used to predict the second transition? I will elaborate on this point below. In general, the early-warnings analysis in Sect. 3 is less clear to me than Sect. 2. I have the impression that the authors present two analyses of a single tipping, and not one analysis of an induced tipping, which somewhat questions the novelty of the approach. At first, I thought that the authors aim to predict the second tipping before the first, or infer what kind of bifurcation to expect. However, after the first examples of cascading tipping it seemed like the approach was to use early warning signals to first predict the first transition and after that predict the second transition, but to do that the concept of cascading tipping is not necessary, since they basically predict two tipping points independent from**

[Figure]

**each other.**

Authors reply:
*We agree with the referee that what is discussed in section 3 is broader than only
the prediction of cascading transitions. The goal of section 3 is to give an overview
of some important metrics during a cascading transition event, to see whether we
can find early warnings of the complete (cascading) event, or to see whether we can
find signals that a first (already happened) transition brings another system closer to
second transition. So the question was indeed not narrowed to (only) predicting the
second tipping before the first. Basically we tried to answer three questions: Can we
predict the complete event (second tipping before the first)? For this we looked at
DCCA and $\rho_{DCCA}$. Can we diagnostically see whether the second tipping is caused
by the first? For this we looked at the ratios of AR1, variance and DFA before and after
the first tipping. How do standard statistical metrics act during the event? For this we
looked at AR1, variance and DFA.*

Changes in text:
We will add additional text to the beginning of section 3 to explain better what the goal
of the section is. We will also rephrase page 2, line 32. We will discuss the limits of
the DCCA and $\rho_{DCCA}$ usage.

4. Comment of the referee:
**In this context, I was also wondering why the external shock that the second
system receives must result from a bifurcation in the leading system. Could it
not also result from other kinds of tipping points, or a sudden step or peak in
forcing like a volcanic eruption or a pulse release of greenhouse gases? Why is
the leading system needed at all when the main aim is to detect if the first shock
will trigger a transition in the following system?**

Authors reply:

*Of course, relating two transitions to each other is not trivial. It might indeed be that a second transition is completely unrelated to the first. If we would like to check the relation between a second transition to a first, we propose to look at the ratio in the autocorrelation, variance and DFA, because apparently (see Tab. 3-4) these are significantly different from events where there is no second transition or when they are unrelated.*

Changes in text:
We will clarify this together with the changes according to the comment 3 above.

5. Comment of the referee:
**The model example in the end (ENSO-AMOC model) is interesting, but its purpose is not clear enough to me. Maybe the authors can clarify what it is that they want to demonstrate exactly and state this clearly in the introduction and draw conclusions using the results they show.**

Authors reply:

*Section 4 is an example of a cascading transition in a physical model. We do not claim that this experiment reproduces reality, but it does reflect that cascading transitions are not a pure mathematical construct, but that they present inside idealized climate models.*

Changes in text:
We will elaborate on the goal of section 4 at the beginning of the section and discuss its implications.

6. Comment of the referee:
**The conclusion section should be extended by a discussion about what questions are answered and what the implications of the results are. What can we do or understand with the approach in this paper that we were not able to do or understand before? What should be done next?**

Authors reply:
*Agreed, the implications can be more explicit. The main new notion is that tipping in one system can lead to tipping in another system through modification of the stability of the state of the latter system, even if both systems are only weakly coupled. An important implication of this is that several climate subsystems may actually be highly vulnerable through their coupling with other subsystems.*

Changes in text:
We will add a paragraph in the revised discussion section to reflect on the main implications of the results in the paper.

7. Comment of the referee:
**The reasons for the choice of methods should be explained better. This is often linked to the problem mentioned above, i.e. the lack of clarity about the aim of the study. Once this aim becomes clearer, it should also be easier to explain why certain methods are applied. Specifically, the choice of statistical indicators needs better justification. I currently do not see what the early warnings approach can add to previous studies. For example, why is DFA used as a warning signal instead of just the autocorrelation? Since autocorrelation is simpler to calculate and more intuitive, I would like to see an argument for the**

**added value of DFA.**

Authors reply:
*The referee makes a fair point that DFA does not add much in the simple systems we look at right now. As we can see in Fig. 3-5 and Tab. 3-4, AR1 and variance mark the slowing down pretty well, even better than DFA in terms of robustness and statistical significance. DFA is argued to be needed when one needs to filter long-range correlations/non-stationarity in data that has a relatively short size with respect to short-range noise, e.g. in Greenland ice core data in Livina & Lenton (2007). For completeness, we added DFA because when one applies these ideas to actual data, DFA might be necessary.*

Changes in text:
We will motivate better why we use DFA, in line with the changes from comment 3.

8. Comment of the referee:
**The statement that "standard quantities not always provide an early warning signal" (page 6, line 26/27) should be backed up with an argument and references, and then it should be explained why DFA can cope with this.**

Authors reply:
*Agreed. 'Standard quantities' is ill-defined in this sentence. As discussed in comment 3 and 7, the DFA argumentation is also elaborated on more.*

Changes in text:
The statement will be changed into 'standard metrics like autocorrelation at lag 1 and variance do not always provide an early warning signal (e.g. in Greenland ice core

data in Livina & Lenton 2007).'

9. Comment of the referee:
**I would actually expect DFA to fail whenever autocorrelation fails, which happens when the system is more complicated than the typical Langevin equation / AR1-process with one fixed time scale. One argument the authors give is that "DFA copes well with non-stationarity". First: What is the explanation for this statement? What is the tradeoff when using DFA (more data needed?). Second: Couldn't one just remove non-stationarity with a high-pass filter (which is what the authors seem to do already) and then use traditional early warning signals? The authors use relatively simple models here, where the parameter can be varied as slowly as necessary to remove non-stationarity (or they could even make long stationary time series for different fixed parameter values). Another argument the authors provide is that DFA captures long-range correlations. But why should one expect such long-range correlations in the simple models the authors use? Can they even exist? So, in a nutshell, why is DFA needed in this paper?**

Authors reply:
*The argumentation for the usage of DFA is found in (among others) in Livina & Lenton 2007 and Peng 1994. Livina & Lenton (2007) argue that DFA filters long-range correlations or non-stationarity better. They apply it on a dataset of Greenland ice cores and argue why degenerate fingerprinting is less applicable (due to the short length of the dataset with respect to the time scale of the non-stationarity).*

Changes in text:
We already refer to these articles in the paper, but in line with comments 3, 7 and 8, we will explain this better.

10. Comment of the referee:

**Then the authors generalise DFA to capture the involvement of several state variables (using DCCA). This could make sense if they were trying to detect something about the coupled system, for instance, which variable is leading, what will happen after tipping 1 and 2. However, the main results seem to consist in predicting tipping 1, and then detecting that the following system has moved closer to a tipping point (by the way, how do we know that there is a second tipping point? The fluctuations could just have changed for another reason.). As far as I can see, DCCA is not needed to do so, an AR1-analysis of each single variable may have sufficed.**

Authors reply:

*As stated in comment 3, we look at multiple statistical aspects of cascading transitions. DCCA is used to see whether the detrended cross-correlation gives any signal of the complete system prior to a cascading transition, which lets it act as an early warning signal. We agree that this can be made more clear, and that DCCA is not adding much in terms of results. However, we think that it does add to the completeness of the statistical description and invites the scientific community to do further research on this.*

Changes in text:
We will add some text in 3.2 to explain why DCCA is included.

11. Comment of the referee:

**The explanation on page 7, lines 26-30 is unclear to me. In what way and to what purpose and why can Pearson's correlation "not be used"? And what is meant with a "one-to-one-relationship" (line 30)?**

Authors reply:

*Not much is predictable about the behaviour of the following system prior to the first transition, as it might be still far away from its bifurcation point. The pure (Pearson's) cross-correlation might therefore be very noisy or even decreasing prior to the first tipping. A 'one-to-one relationship' refers to that the systems correlate well, but this is clearly not necessarily the case. However, we suggest that by looking at long-range correlations (e.g. using DCCA), one might filter out an increase in the long-range cross-correlation prior to the first transition, acting like an early warning signal for cascading transitions. The results are too noisy to interpret, but we invite other researchers to look at this more in-depth.*

Changes in text:
The first part of 3.2 will be adapted as in the reply to comment 10.

12. Comment of the referee:
**Sect. 3.3.1: The authors state several times that DXA and DCCA are sensitive to the segment size and moving window size, but have different values been tried? It would be nice to show how sensitive they are, and what this means for the results. It could help already to just show more runs with different parameter settings.**

Authors reply:
*Indeed, we have tried different values. The results are indeed quite sensitive to seg-ment size (for detrending in DFA/DCCA) and moving window (for running averages).*

Changes in text:

We will a short discussion on the effect of segment size on the results.

13. Comment of the referee:
**In section 3.1 The essential part of degenerate fingerprinting is the projection on the leading EOF in a multivariate system. However, the manuscript skips this part of the method, and therefore, right now, just explains the lag-1 autocorrelation and not degenerate fingerprinting. Could one learn something about a system with cascading tipping points by using degenerate fingerprinting on the multivariate signal?**

Authors reply:
*We agree with the referee. We only focus on the proposed c-propagator as in Livina & Lenton 2007, without explaining the background of degenerate fingerprinting. The c-propagator is reflected by the AR1 coefficient and therefore is an estimator of the degenerate fingerprinting technique.*

Changes in text:
In section 3.1, we will add a few lines on the decay rate of $\kappa$ and the projection on the leading mode.

14. Comment of the referee:
**Sect. 3.3: Why are only the double-fold and fold-Hopf systems tested for the early-warning approach, and not the two systems with a Hopf bifurcation in the leading system? This choice should be explained or the other two examples should be included as well.**

Authors reply:

*We agree that this seems arbitrary. We focus on those two systems because adding more would be repetitive and not adding much, and the choice of these two is because they both start with a clear first transition, making them more illustrative.*

Changes in text:
At the beginning of 3.3, we will add additional text to explain why we focus on the double-fold and fold-Hopf cases.

15. Comment of the referee:
**Sect. 3.3.2, page 10, last paragraph: The oscillation seems to affect the measurement of autocorrelation. I think that one should here measure the auto-correlation of the residuals around a mean oscillation, either by subtracting this mean cycle somehow, or by defining a period and working with Poincare sections (snapshots after each period). Otherwise the result would probably be meaningless.**

Authors reply:
*This is a good suggestion. We will subtract the mean oscillation.*

Changes in text:
We will change Fig. 4 and its discussion by removing the mean oscillation.

16. Comment of the referee:
**In both figure 1 and figure 2, the choice for the coupling of the two subsystems seems to be arbitrary. These choices could be explained better to make it more understandable for the reader. For example, one can shift the two systems versus each other (by varying parameter gamma1), such that the two tippings**

**are well separated, or that they are really intertwined (one tipping inducing the other immediately). How would the stability landscape then look like, and what would we see in early-warning signals? I was also wondering why the values of gamma1 have been chosen in a way that gamma1 is 0 for the double-fold, <0 for the Fold-Hopf, and double-Hopf, and >0 for the Hopf-fold. Conceptually it would make a difference if the second tipping is triggered by the changing parameter or a direct consequence of the first tipping. It seems that the latter is always the case here for all parameter choices? This should be made clear from the beginning (as I mentioned above).**

Authors reply:
*For the discussion on the separation of the transitions, we refer to our answer to comment 2. The parameter settings are chosen such that a cascading event is produced. As specified in Tab. 1-2, there are boundaries within which one can vary the parameters (for some values, there will not be a second tipping). Within these boundaries, one could vary $\gamma_1$ to make the transient of the following system progress slower or faster towards the new equilibrium, but the stability landscape would remain the same as in Fig. 1 (note that this landscape has phi on the horizontal axis). And again; the second tipping is triggered by the first tipping, and therefore indirectly by the forcing parameter $\phi$.*

Changes in text:
None.

17. Comment of the referee:
**Probably related to this point: In figure 2 it seems to be the case that the leading system tips before the following system, whereas the bifurcation plots seem to indicate that this happens at the same time. Where does this time delay come**

[Figure]

**from?**

Authors reply:
*See our reply on comment 2 and 16.*

Changes in text:
None additional.

18. Comment of the referee:
**Similarly, a time delay can be seen in the Fold-Hopf system (Fig. 2b), while Fig. 1b would make me expect a discontinuous jump from a stationary solution to a cycle with some non-zero amplitude.**

Authors reply:
*It takes some time before the cycle arises. Strictly speaking, the following system in 2b would actually remain near the unstable equilibrium for a long time (if there would be no numerical imperfections), as the system is deterministic. The latter can, however, be stated more clearly. In any case, there would be no jump to a non-zero amplitude cycle in the transient. This would always be gradual.*

Changes in text:
We will add a sentence to say that in a pure deterministic case, the following system in the Fold-Hopf case would remain near the unstable equilibrium for a long time.

19. Comment of the referee:
**Also, according to Table 2, the control parameter Phi increases linearly with**

time, but I do not see any change in state (or the amplitude of its oscillations) in Fig. 2, and on page 5, last paragraph, it is mentioned that at some point the amplitude would jump to a large value when both equilibria are accessed, but this is not seen in the Figure.

Authors reply:

*Assuming the referee is talking about the Hopf-Fold, a change in state in the leading system is visible in 2c in the fact that the black/grey lines start oscillating as soon as $\phi$ is large enough (i.e., when the lines go from dashed to solid, reaching the critical value). In the following system, the change in state is also visible in 2c, by observing the sharp increase in the red curve around t = 130.*

Changes in text:
None.

20. Comment of the referee:

**In this context, Fig. 1 and 2 appear contradictory to me. This point is actually a crucial one because the period between the two tippings is used to detect early warning signals for the second tipping. How can it even be that there is enough time to detect them, when the system is already in the process of tipping? Here it looks like the second tipping is actually not caused directly by the first, but by the changing control parameter (in contrast to the impression I got in the previous section).**

Authors reply:

*The second tipping is caused by the first, and the first is caused by the changing control parameter. You can directly see this in the equations, where the leading system's state*

*variables act directly as bifurcation parameters for the following system. An example is the double fold case (Eqn. 13), where gamma modulates the (bi-)stability of the following system, and is dependent on $X$, not (directly) on $\phi$. However, the equilibrium of $X$ is determined by $\phi$, which is why for a varying $\phi$, $X$ might transition towards a completely other state, which affects $\gamma(X)$ so dramatically that it might affect the stability of $Y$. To summarise: $\phi$ indirectly affects the transition in the following system, but always through the transition in the leading system. Concerning the detection time: Fig. 1 and 2 are for deterministic systems and therefore not really about detection, as for detection, a lot of information is gained from its behaviour around its equilibrium (in terms of noise and recovery from perturbations). Also note that in deterministic systems, cascading transition per definition does not have a state between the two tipping events, where the following system is stable (as in that case, the following system would remain stable; there is no noise to change that). Early warnings are analysed in section 3 (Fig. 3-5), where we look at stochastic systems. In contrast with the systems in Fig. 1-2, in section 3, there is time between the first and second tipping (called the following transition period in this paper) where the following system is stable, but close to its bifurcation point (e.g., resulting in high AR1).*

Changes in text:
We will add additional text to the beginning of section 2 to emphasise the fact that the systems presented there are deterministic and explain what we replied to this comment.

21. Comment of the referee:
**If it is a real cascade (tipping 2 directly induced by tipping 1), wouldn't the system's state suddenly be very far from equilibrium after tipping 1. Can early warnings even be expected in this situation (mind they sample the equilibrium when the state fluctuates around it)?**

Authors reply:

*The idea here is that there are two different systems; a leading system and a following system. The first tipping alters background conditions such that the following system comes closer to its bifurcation point. It might be that the referee means that the cascade might not be a consequence of crossing bifurcation points, but rather a cascade as a consequence of a strong perturbation, which is unrelated to changing equilibria or stability in the leading system. That would indeed bring the leading system's state far away from its equilibrium, and depending on its recovery rate, might bring it back to the pre-existing stable equilibrium. During this phase, it might also affect the following system, if that system is coupled in the right way to the leading system. Perturbations are (without any pre-knowledge on the source of the perturbations) unpredictable, but the time between the first and second tipping might give warning indicators about a second tipping.*

Changes in text:
None.

22. Comment of the referee:

**Moreover, I imagine that the relative time scale between the systems matters (controlled by the different coefficients in the equations). For example, in case of the Hopf-fold system, it would matter how fast and how large the oscillation in the leading system is compared to the following system's response time.**

Authors reply:

*The relative time scale between the systems indeed is a factor to take into account. For now, we used relatively equal time scales. Of course, if the following system has*

*a very short time scale, and the leading system a rather long time scale, the following system might still transit to a new state as a consequence of the leading's transition, but is hard to find any time in between the transitions, and in real data it might be hard to distinguish the sequence.*

Changes in text:
None.

23. Comment of the referee:
**So why has this particular coupling been chosen for the paper, and how representative is that for the climate system?**

Authors reply:
*The linear coupling is based on the thermal wind balance, where the wind stress adjusts to the changes in meridional temperature gradient. So the coupling formulation has a physical basis.*

Changes in text:
We will motivate the coupling better in the revised section 4.

24. Comment of the referee:
**The climate model (coupled ENSO and AMOC) seems very interesting. However, it is not completely clear to me what point exactly the authors want to make by showing it. Sect. 4.3 is very short and I don't really understand its purpose. In the conclusion section and in the abstract it seems to be argued that it illustrates that cascading tipping can occur in climate models, but as this is already known according to the introduction, and given that the model has been designed like**

**this on purpose, what new information does this model provide?**

Authors reply:
*Agreed; this is indeed less clear from the text.*

Changes in text:
We will add a short motivation in the beginning of section 4 why this model is chosen.

25. Comment of the referee:
**Also, it should be more clearly explained how the two existing models have been coupled. I found it difficult at first to identify the common variables in the models that were linked. More precise wording might help (e.g. "through influence of the wind stress" - influence on what?, "in the original model" - which model?). It seems that the authors introduce an equation for the wind stress $\tau$ which links $\tau$ from the ENSO model to the temperatures from Stommel's model? Then one could say so from the start, followed by the details.**

Authors reply:
*The sentence containing 'through influence of the wind stress' is indeed unclear. Section 4.2 gives a technical overview of the complete coupled model. The coupling itself is part of that and the introduction and explanation of the coupling appears sufficient.*

Changes in text:
We will correct the mentioned sentence.

26. Comment of the referee:
**The model seems to be a representation of the Fold-Hopf case above? This should be explicitly stated from the beginning.**

Authors reply:
*We agree.*

Changes in text:
At the beginning of section 4, we will add that this is an example of the Fold-Hopf case.

27. Comment of the referee:
**- Why have the authors not done an early-warning analysis with this AMOC-ENSO model? This would be a natural step to do after the generic models above.**

Authors reply:
*This is indeed something we have thought about. The largest problem is that the model is deterministic. Some stochasticity would be needed to be build into the model, which would require a thorough sensitivity analysis and needs to be done with care as the pre-existing models were not designed for that. This is beyond the scope of this paper, if even possible. Moreover, it should be noted that this is not the aim of section 4, which is illustrating an application of cascading transitions in physics, implying the possibility of these events in physical systems.*

Changes in text:
None.

28. Comment of the referee:

**The authors use data from a complex model to tune their conceptual model. What can we learn from that data directly about predicting each tipping, or the coupling (or whatever the authors aim to do)? Could one apply a statistical analysis and infer something about that model from the data?**

Authors reply:

*We only use the complex model data to back up the ideas we have for (a) the coupling (relation between $\tau_{ext}$ and meridional temperature) and (b) what happens to El-Niño when the AMOC collapses. Trying to infer predictions/early warning elements in this data has a number of problems. A first problem is that the first transition is caused by a large perturbation, and no critical slowing down has happened prior to that. So the period before the first transition is useless in the scope of cascading tipping prediction. The second problem is that we are unsure (and we also do not claim otherwise in the paper), whether a (stochastic) Hopf bifurcation can be found or is crossed in Pacific Equatorial SSTs in the hosing runs, like in our simplified model. More research is needed for that.*

Changes in text:

We will add a few lines on the limitations of the data from the complex model in the revised section 4.

29. Comment of the referee:

**Minor comments What I find most interesting is the analysis of the coupled deterministic systems, e.g. in Fig. 1. A very interesting aspect is the occurrence of intermediate (in terms of the state variable) stable states which are inaccessible when varying the control parameter. It seems that only noise can bring the system on these branches. This aspect is however not discussed in the paper. It**

**is of course up to the authors if they want to go into this, but I would recommend them to at least comment on these hidden states, which I personally find more novel and exciting than the early-warning part of the paper. Could there be such hidden stable states in the climate system and how can they be found?**

Authors reply:
*Thank you for mentioning this. The occurrence of these in-between states (called following transition period, FTP in section 3) in stochastic systems is indeed interesting.*

Changes in text:
We will add a few sentences about the FTP, explaining it more elaborately, in section 3.3.1.

30. Comment of the referee:
**Section 2.1 + Figure 1: It took me quite some time to understand what is going on. It could be helpful to create an X-Y bifurcation plot in addition to the phi-X plots and phi-Y plots that are shown already (to see how each system behaves in isolation). More emphasis can be put on explaining this figure, because this in itself is already an interesting result. The authors might even think of making an animation as extra material, to show how the subplots relate to each other. Also, it could be nice to show how figure 2 relates to figure 1.**

Authors reply:
*This is indeed confusing, see also the reply on comment 2.*

Changes in text:
We will add a sentence in the beginning of section 2, explaining Fig. 1 in more detail.

We will also add extra panels in Fig. 1 to see the bifurcation diagram of the following system with respect to the coupling and to emphasize the separation between the systems .

31. Comment of the referee:
**- Several different names are sometimes used for the same thing, at least for the fold bifurcation (fold / back-to-back / back-to-back saddle-node). I had never come across the term back-to-back before. Is one term a subset of another? The authors should clarify this and unify the language.**

Authors reply:
*We indeed used all those terms interchangeably. The back-to-back saddle-node consists of two saddle-nodes at ends of two stable equilibrium branches, connected by a common unstable equilibrium branch.*

Changes in text:
We will explain the prefix 'back-to-back' in section 2 and use only the terms saddle-node bifurcation and fold in the revised paper.

32. Comment of the referee:
**- Some of the references are a bit outdated (e.g. Kutzbach 1996 on page 2; a lot has happened since then), or could be a bit more specific. Page 2: Scheffer 2009 is a review of some of the earlier papers like Held 2004, some of which are cited later; Peng 1994 is not about predicting tipping points. Also, note that there are papers from the 80ies dealing with statistical precursors already, e.g. by Wiesenfeld, 1984.**

Authors reply:
*Although perhaps outdated, Kutzbach 1996 is illustrative for what is said in our paper about the desertification of the Sahel region. We agree that Scheffer 2009 is a review paper, but it combines various simple metrics together. Held 2004 is cited multiple times when we specifically mention degenerate fingerprinting, about which Scheffer 2009 does not go into detail. Peng 1994 lays the foundation of DFA to detect long-range signals, and therefore should (only) be cited when we talk about DFA, which we do. Wiesenfeld 1984 is specifically about period-doublings.*

Changes in text:
None.

33. Comment of the referee:
**page 1 (lines 17ff): Lenton et al. 2008 do not show evidence that there are tipping points in the climate system (though the paper is often cited in that way), so this paragraph should be formulated more cautiously.**

Authors reply:
*We agree.*

Changes in text:
Part of the sentence will be changed to "... Lenton et al. (2008) give an overview of these."

34. Comment of the referee:
**Also, the vegetation states found by Hirota et al. are purely ecological phenomena, and do not imply any tipping points in the climate.**

Authors reply:
*Hirota et al. 2011 show various equilibrium states of tree cover as modulated by precipitation, and also discusses transitions between these states, and related hysteresis effects and bifurcations. Although the paper focuses on the interaction between vegetation and precipitation, the results are illustrative of tipping elements in (a subsystem of) the climate system.*

Changes in text:
None.

35. Comment of the referee:
**- page 6, line 18: "close to critical transition" (2x), should be "close to a critical transition".**

Authors reply:
*Agreed.*

Changes in text:
This will be changed accordingly.

36. Comment of the referee:
**- page 10, line 17/18: "as it is no critical transition": why not? And what is a critical transition?**

Authors reply:

*Agreed that this needs to be rephrased.*

Changes in text:
The sentence will be rewritten.

37. Comment of the referee:
**- In Fig. 8, I found it confusing that the labels are not next to the vertical axes but inside the figure. I do understand that this is consistent with the previous figures, so I don't have strong feelings about this.**

Authors reply:
*Understandable, and a matter of choice. As we have had no other comments about this, we will leave it as it is.*

Changes in text:
None.